# WEAKLY SUPERVISED EXPLAINABLE PHRASAL REASONING WITH NEURAL FUZZY LOGIC

**Zijun Wu**[*1], **Zi Xuan Zhang**[*1], **Atharva Naik**[†2], **Zhijian Mei**[1], **Mauajama Firdaus**[1], **Lili Mou**[1]

[1]Dept. Computing Science & Alberta Machine Intelligence Institute (Amii), University of Alberta
[2]Carnegie Mellon University
{zijun4, zixuan7, zimei1}@ualberta.ca, arnaik@cs.cmu.edu,
{mauzama.03, doublepower.mou}@gmail.com
[*]Equal contribution, [†]Work done during the internship at UofA/Amii

## ABSTRACT

Natural language inference (NLI) aims to determine the logical relationship between two sentences, such as `Entailment`, `Contradiction`, and `Neutral`. In recent years, deep learning models have become a prevailing approach to NLI, but they lack interpretability and explainability. In this work, we address the explainability of NLI by weakly supervised logical reasoning, and propose an Explainable Phrasal Reasoning (EPR) approach. Our model first detects phrases as the semantic unit and aligns corresponding phrases in the two sentences. Then, the model predicts the NLI label for the aligned phrases, and induces the sentence label by fuzzy logic formulas. Our EPR is almost everywhere differentiable and thus the system can be trained end to end. In this way, we are able to provide explicit explanations of phrasal logical relationships in a weakly supervised manner. We further show that such reasoning results help textual explanation generation.[1]

## 1 INTRODUCTION

Natural language inference (NLI) aims to determine the logical relationship between two sentences (called a *premise* and a *hypothesis*), and target labels include `Entailment`, `Contradiction`, and `Neutral` (Bowman et al., 2015; MacCartney & Manning, 2008). Figure 1 gives an example, where the hypothesis contradicts the premise. NLI is important to natural language processing, because it involves logical reasoning and is a key problem in artificial intelligence. Previous work shows that NLI can be used in various downstream tasks, such as information retrieval (Karpukhin et al., 2020) and text summarization (Liu & Lapata, 2019).

In recent years, deep learning has become a prevailing approach to NLI (Bowman et al., 2015; Mou et al., 2016; Wang & Jiang, 2016; Yoon et al., 2018). Especially, pretrained language models with the Transformer architecture (Vaswani et al., 2017) achieve state-of-the-art performance for the NLI task (Radford et al., 2018; Zhang et al., 2020). However, such deep learning models are black-box machinery and lack interpretability. In real applications, it is important to understand how these models make decisions (Rudin, 2019).

Several studies have addressed the explainability of NLI models. Camburu et al. (2018) generate a textual explanation by sequence-to-sequence supervised learning, in addition to NLI classification; such an approach is multi-task learning of text classification and generation, which does not perform reasoning itself. MacCartney et al. (2008) propose a scoring model to align related phrases; Parikh et al. (2016) and Jiang et al. (2021) propose to obtain alignment by attention mechanisms. However, they only provide correlation information, instead of logical reasoning. Other work incorporates upward and downward monotonicity entailment reasoning for NLI (Hu et al., 2020; Chen et al., 2021), but these approaches are based on hand-crafted rules (e.g., *every* downward entailing *some*) and are restricted to `Entailment` only; they cannot handle `Contradiction` or `Neutral`.

In this work, we address the explainability for NLI by weakly supervised phrasal logical reasoning. Our goal is to explain NLI predictions with phrasal logical relationships between the premise and

---

[1]Code and resources available at https://github.com/MANGA-UOFA/EPR

hypothesis. Intuitively, an NLI system with an explainable reasoning mechanism should be equipped with the following functionalities:

1. The system should be able to detect corresponding phrases and tell their logical relationship, e.g., *several men* contradicting *one man*, but *pull in a fishing net* entailing *holding the net* (Figure 1).
2. The system should be able to induce sentence labels from phrasal reasoning. In the example, the two sentences are contradictory because there exists one contradictory phrase pair.
3. More importantly, such reasoning should be trained in a weakly supervised manner, i.e., the phrase-level predictions are trained from sentence labels only. Otherwise, the reasoning mechanism degrades to multi-task learning, which requires massive fine-grained human annotations.

To this end, we propose an Explainable Phrasal Reasoning (EPR) approach to the NLI task. Our model obtains phrases as semantic units, and aligns corresponding phrases by embedding similarity. Then, we predict the NLI labels (namely, `Entailment`, `Contradiction`, and `Neutral`) for the aligned phrases. Finally, we propose to induce the sentence-level label from phrasal labels in a fuzzy logic manner (Zadeh, 1988; 1996). Our model is differentiable, and the phrasal reasoning component can be trained

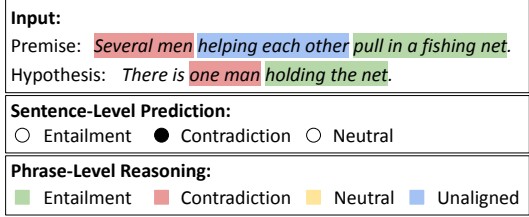

Figure 1: The natural language inference (NLI) task and desired phrasal reasoning.

with the weak supervision of sentence NLI labels. In this way, our EPR approach satisfies all the desired properties mentioned above.

In our experiments, we developed a comprehensive methodology (data annotation and evaluation metrics) to quantitatively evaluate phrasal reasoning performance, which has not been accomplished in previous work. We extend previous studies and obtain plausible baseline models. Results show that our EPR yields a much more meaningful explanation regarding $F$ scores against human annotation.

To further demonstrate the quality of extracted phrasal relationships, we feed them to a textual explanation model. Results show that our EPR reasoning leads to an improvement of 2 points in BLEU scores, achieving a new state of the art on the e-SNLI dataset (Camburu et al., 2018).

Our contributions are summarized as follows:

1. We formulate a phrasal reasoning task for natural language inference (NLI), addressing the interpretability of neural models.
2. We propose an EPR model that induces sentence-level NLI labels from explicit phrasal logical labels by neural fuzzy logic. EPR is able to perform reasoning in a weakly supervised way.
3. We annotated phrasal logical labels and designed a set of metrics to evaluate phrasal reasoning. We further use our reasoning results to improve textual explanation generation. Our code and annotated data are released for future studies.

To the best of our knowledge, we are the first to develop a weakly supervised phrasal reasoning model for the NLI task.

## 2 RELATED WORK

**Natural Language Inference.** MacCartney & Manning (2009) propose seven natural logic relations in addition to `Entailment`, `Contradiction`, and `Neutral`. MacCartney & Manning (2007) also distinguish upward entailment (*every mammal* upward entailing *some mammal*) and downward entailment (*every mammal* downward entailing *every dog*) as different categories. Manually designed lexicons and rules are used to interpret `Entailment` in a finer-grained manner, such as downward and upward entailment (Hu et al., 2020; Chen et al., 2021). Feng et al. (2020) apply such natural logic to NLI reasoning at the word level; however, our experiments will show that their word-level treatment is not an appropriate granularity, and they fail to achieve meaningful reasoning performance.

The above reasoning schema focuses more on the quantifiers of first-order logic (Beltagy et al., 2016). However, the SNLI dataset (Bowman et al., 2015) we use only contains less than 5% samples with explicit quantifiers, and the seven-category schema complicates reasoning in the weakly supervised

setting. Instead, we adopt three-category NLI labels following the SNLI dataset. Our focus is entity-based reasoning, and the treatment of quantifiers is absorbed into phrases.

We also notice that previous work lacks explicit evaluation on the reasoning performance for NLI. For example, the SNLI dataset only provides sentence-level labels. The HELP (Yanaka et al., 2019a) and MED (Yanaka et al., 2019b) datasets concern monotonicity inference problems, where the label is also at the sentence level; they only consider Entailment, ignoring Contradiction and Neutral. Thus, we propose a comprehensive framework for the evaluation of NLI reasoning.

**e-SNLI.** Camburu et al. (2018) propose the e-SNLI task of textual explanation generation and use LSTM as a baseline. Kumar & Talukdar (2020) propose the NILE approach, using multiple decoders to generate explanations for all E, C, and N labels, and then predicting which to be selected. Zhao & Vydiswaran (2021) propose the LIREx approach, using additionally annotated rationales for explanation generation. Narang et al. (2020) finetune T5 with multiple explanation generation tasks. Although these systems can generate explanations, the nature of such finetuning approaches renders the explanation generator *per se* unexplainable. By contrast, we design a textual explanation generation model that utilizes our EPR's phrasal reasoning, obtained in a weakly supervised manner.

**Neuro-Symbolic Approaches.** In recent years, neuro-symbolic approaches have attracted increasing interest in the AI and NLP communities for interpreting deep learning models. Typically, these approaches are trained by reinforcement learning or its relaxation, such as attention and Gumbel-softmax (Jang et al., 2017), to reason about certain latent structures in a downstream task.

For example, Lei et al. (2016) and Liu et al. (2018) extract key phrases or sentences for a text classification task. Lu et al. (2018) extract entities and relations for document understanding. Liang et al. (2017) and Mou et al. (2017) perform SQL-like execution based on input text for semantic parsing. Xiong et al. (2017) hop over a knowledge graph for reasoning the relationships between entities. Li et al. (2019) and Deshmukh et al. (2021) model symbolic actions for unsupervised syntactic structure induction. In the vision domain, Mao et al. (2019) propose a neuro-symbolic approach to learn visual concepts. Our work addresses logical reasoning for the NLI task, which is not tackled in previous neuro-symbolic studies.

**Fuzzy Logic.** Fuzzy logic (Zadeh, 1988; 1996) models an assertion and performs logic calculation with probability. For example, a quantifier (e.g., "most") and assertion (e.g., "ill") are modeled by a score in $(0, 1)$; the score of a conjunction $s(x_1 \wedge x_2)$ is the product of $s(x_1)$ and $s(x_2)$. In old-school fuzzy logic studies, the mapping from language to the score is usually given by human-defined heuristics (Zadeh, 1988; Nozaki et al., 1997), and may not be suited to the task of interest. By contrast, we train neural networks to predict the probability of phrasal logical relations, and induce the sentence NLI label by fuzzy logic formulas. Thus, our approach takes advantage of both worlds of symbolism and connectionism. Mahabadi et al. (2020) apply fuzzy logic formulas to replace multi-layer perceptrons for NLI. But they are unable to provide expressive reasoning because their fuzzy logic works on sentence features. Our work is inspired by Mahabadi et al. (2020). However, we propose to apply fuzzy logic to the detected and aligned phrases, enabling our approach to provide reasoning in a symbolic (i.e., expressive) way. We develop our own fuzzy logic formulas, which are also different from Mahabadi et al. (2020).

## 3 OUR EPR APPROACH

In this section, we describe our EPR approach in detail, also shown in Figure 2. It has three main components: phrase detection and alignment, phrasal NLI prediction, and sentence label induction.

**Phrase Detection and Alignment.** In NLI, a data point consists of two sentences, a premise and a hypothesis. We first extract content phrases from both input sentences by rules and heuristics. For example, "[AUX] + [NOT] + VERB + [RP]" is treated as a verb phrase. Full details are presented in Appendix A.1. Compared with the word level (Parikh et al., 2016; Feng et al., 2020), a phrase is a more meaningful semantic unit for logical reasoning.

We then align corresponding phrases in the two sentences based on cosine similarity. Let $P = (p_1, \cdots, p_M)$ and $H = (h_1, \cdots, h_N)$ be the premise and hypothesis, respectively, where $p_m$ and $h_n$ are extracted phrases. We apply Sentence-BERT (Reimers & Gurevych, 2019) to each individual phrase and obtain the local phrase embeddings by $\boldsymbol{p}_m^{(L)} = \text{SBERT}(p_m), \boldsymbol{h}_n^{(L)} = \text{SBERT}(h_n)$. We

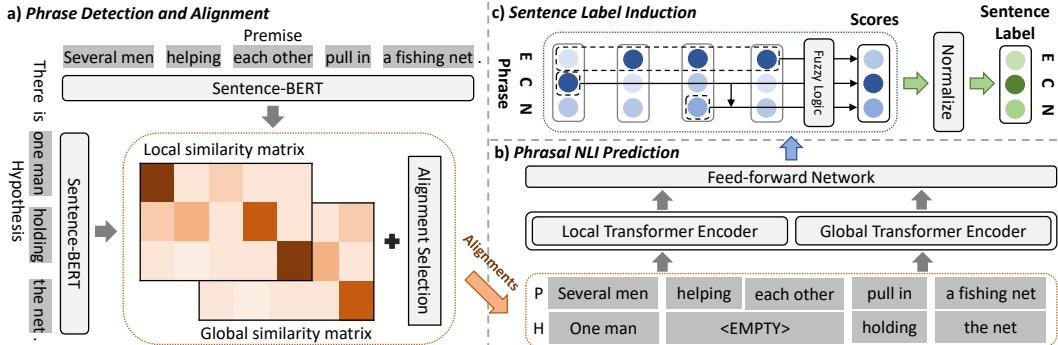

Figure 2: An overview of our Explainable Phrasal Reasoning (EPR) model.

Table 1: An example showing the importance of handling unaligned phrases (in highlight).

| Premise | People are shopping for fruit. | People are shopping for fruit in the market . |
|---|---|---|
| Hypothesis | People are shopping for fruit in the market . | People are shopping for fruit. |
| Sentence NLI | [ ] Entailment  [ ] Contradiction [✓] Neutral | [✓] Entailment  [ ] Contradiction [ ]  Neutral |

also apply Sentence-BERT to the entire premise and hypothesis sentences to obtain the global phrase embeddings $\boldsymbol{p}_m^{(G)}$ and $\boldsymbol{h}_n^{(G)}$ by mean-pooling the features of the words in the phrase. The phrase similarity is given by

$$\text{sim}(\text{p}_m, \text{h}_n) = \gamma \cos(\boldsymbol{p}_m^{(G)}, \boldsymbol{h}_n^{(G)}) + (1 - \gamma) \cos(\boldsymbol{p}_m^{(L)}, \boldsymbol{h}_n^{(L)}) \tag{1}$$

where $\gamma$ is a hyperparameter balancing the lexical and contextual representations of a phrase (Hewitt & Manning, 2019). It is noted that Sentence-BERT is finetuned on paraphrase datasets, and thus is more suitable for phrasal similarity matching than pretrained language models (Devlin et al., 2019).

We obtain phrase alignment between the premise and hypothesis in a heuristic way. For every phrase $\text{p}_m$ in the premise, we look for the most similar phrase $\text{h}_n$ from the hypothesis by

$$n = \text{argmax}_{n'} \text{sim}(\boldsymbol{p}_m, \boldsymbol{h}_{n'}) \tag{2}$$

Likewise, for every phrase $\text{h}_n$ in the hypothesis, we look for the most similar phrase $\text{p}_m$ from the premise. A phrase pair $(\text{p}_m, \text{h}_n)$ is considered to be aligned if $\text{h}_n$ is selected as the closest phrase to $\text{p}_m$, and $\text{p}_m$ is the closest to $\text{h}_n$. Such hard alignment differs from commonly used soft attention-based approaches (Parikh et al., 2016). Our alignment method can ensure the quality of phrase alignment, and more importantly, leave other phrases unaligned (e.g., *helping each other* in Figure 1), which are common in the NLI task. The process is illustrated in Figure 2a.

**Phrasal NLI Prediction.** Our model then predicts the logical relationship of an aligned phrase pair $(\boldsymbol{p}, \boldsymbol{h})$ among three target labels: `Entailment`, `Contradiction`, and `Neutral`. While previous work (Feng et al., 2020) identifies finer-grained labels for NLI, we do not follow their categorization, because it complicates the reasoning process and makes weakly supervised training more difficult. Instead, we adopt a three-way phrasal classification, which is consistent with sentence NLI labels.

We represent a phrase, say, p in the premise, by a vector embedding, and we consider two types of features: a local feature $\boldsymbol{p}^{(L)}$ and a global feature $\boldsymbol{p}^{(G)}$, re-used from the phrase alignment component. They are concatenated as the phrase representation $\boldsymbol{p} = [\boldsymbol{p}^{(L)}; \boldsymbol{p}^{(G)}]$. Likewise, the phrase representation for a hypothesis phrase $\boldsymbol{h}$ is obtained in a similar way. Intuitively, local features force the model to perform reasoning in a serious manner, but global features are important to sentence-level prediction. Such intuition is also verified in an ablation study (§ 4.2).

Then, we use a neural network to predict the phrasal NLI label (`Entailment`, `Contradiction`, and `Neutral`). This is given by the standard heuristic matching (Mou et al., 2016) based on phrase embeddings, followed by a multi-layer perceptron (MLP) and a three-way softmax layer:

$$[P_{\text{phrase}}(\text{E}|\text{p}, \text{h}); P_{\text{phrase}}(\text{C}|\text{p}, \text{h}); P_{\text{phrase}}(\text{N}|\text{p}, \text{h})] = \text{softmax}(\text{MLP}([\boldsymbol{p}; \boldsymbol{h}; |\boldsymbol{p} - \boldsymbol{h}|; \boldsymbol{p} \circ \boldsymbol{h}])) \tag{3}$$

where $\circ$ is the element-wise product, and the semicolon refers to column vector concatenation. E, C, and N refer to the `Entailment`, `Contradiction`, and `Neutral` labels, respectively.

It should be mentioned that a phrase may be unaligned, but plays an important role in sentence-level NLI prediction, as shown in Table 1. Thus, we would like to predict phrasal NLI labels for unaligned

phrases as well, but pair them with a special token ($p_{\langle \text{EMPTY} \rangle}$ or $h_{\langle \text{EMPTY} \rangle}$), whose embedding is randomly initialized and learned by back-propagation.

**Sentence Label Induction.** We observe the sentence NLI label can be logically induced from phrasal NLI labels. Based on the definition of the NLI task, we develop the following induction rules.

Entailment Rule: According to Bowman et al. (2015), a premise entailing a hypothesis means that, if the premise is true, then the hypothesis must be true. We find that this can be oftentimes transformed into phrasal relationships: a premise entails the hypothesis if all paired phrases have the label `Entailment`.

Let $\{(p_k, h_k)\}_{k=1}^{K} \bigcup \{(p_k, h_k)\}_{k=K+1}^{K'}$ be all phrase pairs. For $k = 1, \cdots, K$, they are aligned phrases; for $k = K + 1, \cdots, K'$, they are unaligned phrases paired with the special token, i.e., $p_k = p_{\langle \text{EMPTY} \rangle}$ or $h_k = h_{\langle \text{EMPTY} \rangle}$. Then, we induce a sentence-level `Entailment` score by

$$S_{\text{sentence}}(\mathsf{E}|\text{P}, \text{H}) = \big[ \prod\nolimits_{k=1}^{K'} P_{\text{phrase}}(\mathsf{E}|p_k, h_k) \big]^{\frac{1}{K'}} \tag{4}$$

This works in a fuzzy logic fashion (Zadeh, 1988; 1996), deciding whether the sentence-level label should be `Entailment` considering the average of phrasal predictions.[2] Here, we use the geometric mean, because it is biased towards low scores, i.e., if there exists one phrase pair with a low `Entailment` score, then the chance of sentence label being `Entailment` is also low. Unaligned pairs should be considered in Eq. (4), because an unaligned phrase may indicate `Entailment`, shown in the second example of Table 1. Notice that the resulting value $S_{\text{sentence}}(\mathsf{E}|\text{P}, \text{H})$ is not normalized with respect to `Contradiction` and `Neutral`; thus, we call it a score (instead of a probability), which will be normalized afterwards.

Contradiction Rule: Two sentences are contradictory if there exists (at least) one paired phrase labeled as `Contradiction`. The fuzzy logic version of this induction rule is given by

$$S_{\text{sentence}}(\mathsf{C}|\text{P}, \text{H}) = \max_{k=1,\cdots,K} P_{\text{phrase}}(\mathsf{C}|p_k, h_k) \tag{5}$$

Here, the `max` operator is used in the induction, because the contradiction rule is an existential statement, i.e., *there exist(s)* $\cdots$. Also, unaligned phrases are excluded in calculating the sentence-level `Contradiction` score, because an unaligned phrase indicates the corresponding information is missing in the other sentence and it cannot be `Contradiction` (recall examples in Table 1).

Rule for Neutral: Two sentences are neutral if there exists (at least) one `neutral` phrase pair, but there does not exist any contradictory phrase pair. The fuzzy logic formula is

$$S_{\text{sentence}}(\mathsf{N}|\text{P}, \text{H}) = \big[ \max_{k=1,\cdots,K'} P_{\text{phrase}}(\mathsf{N}|p_k, h_k) \big] \cdot \big[ 1 - S_{\text{sentence}}(\mathsf{C}|\text{P}, \text{H}) \big] \tag{6}$$

The first factor determines whether there exists a `Neutral` phrase pair (including unaligned phrases, illustrated in the first example in Table 1). The second factor evaluates the negation of "at least one contradictory phrase," as suggested in the second clause of the Rule for Neutral.

Finally, we normalize the scores into probabilities by dividing the sum, since all the scores are already positive. This is given by

$$P_{\text{sentence}}(\mathsf{L}|\cdot) = \frac{1}{Z} S_{\text{sentence}}(\mathsf{L}|\cdot) \tag{7}$$

where $\mathsf{L} \in \{\mathsf{E}, \mathsf{C}, \mathsf{N}\}$, and $Z = S_{\text{sentence}}(\mathsf{E}|\cdot) + S_{\text{sentence}}(\mathsf{C}|\cdot) + S_{\text{sentence}}(\mathsf{N}|\cdot)$ is the normalizing factor.

**Training and Inference.** We use cross-entropy loss to train our EPR model by minimizing $-\log P_{\text{sentence}}(\mathsf{t}|\cdot)$, where $\mathsf{t} \in \{\mathsf{E}, \mathsf{C}, \mathsf{N}\}$ is the groundtruth sentence-level label.

Our underlying logical reasoning component can be trained end-to-end by back-propagation in a weakly supervised manner, because the fuzzy logic rules are almost everywhere differentiable. Although the `max` operators in Eqs. (5) and (6) may not be differentiable at certain points, they are common in max-margin learning and the rectified linear unit (ReLU) activation functions, and do not cause trouble in back-propagation.

Once our EPR model is trained, we can obtain both phrasal and sentence-level labels. This is accomplished by performing `argmax` on the predicted probabilities (3) and (7), respectively.

---

[2]In traditional fuzzy logic, the conjunction is given by probability product (Zadeh, 1988). We find that this gives a too small `Entailment` score compared with `Contradiction` and `Neutral` scores, causing difficulties in end-to-end training. Thus, we take the geometric mean and maintain all the scores in the same magnitude.

**Improving Textual Explanation.** Camburu et al. (2018) annotated a dataset to address NLI interpretability by generating an explanation sentence. For the example in Figure 1, the reference explanation is "There cannot be one man and several men at same time."

In this part, we apply the predicted phrasal logical relationships to textual explanation generation and examine whether our EPR's output can help a downstream task. Figure 3 shows the overview of our textual explanation generator. We concatenate the premise and hypothesis in the form of "*Premise : · · · Hypothesis : · · ·*," and feed it to a standard Transformer encoder (Vaswani et al., 2017).

We utilize the phrase pairs and our predicted phrasal labels as factual knowledge to enhance the decoder. Specifically, our EPR model yields a set of tuples $\{(\mathrm{p}_k, \mathrm{h}_k, \mathrm{l}_k)\}_{k=1}^K$ for a sample, where $\mathrm{l}_k \in \{\mathsf{E}, \mathsf{N}, \mathsf{C}\}$ is the predicted phrasal label for the aligned phrases, $\mathrm{p}_k$ and $\mathrm{h}_k$. We embed phrases by Sentence-BERT: $\boldsymbol{p}^{(L)}$ and $\boldsymbol{h}^{(L)}$; the phrasal label is represented by a one-hot vector $\boldsymbol{l}_k = \mathrm{onehot}(\mathrm{l}_k)$. They are concatenated as a vector $\boldsymbol{m}_k = [\boldsymbol{p}_k; \boldsymbol{h}_k; \boldsymbol{l}_k]$. We compose the vectors as a factual memory matrix $\mathbf{M} = [\boldsymbol{m}_1^\top; \cdots; \boldsymbol{m}_K^\top] \in \mathbb{R}^{K \times d}$, where $d$ is the dimension of $\boldsymbol{m}_k$.

Our decoder follows a standard Transformer architecture (Vaswani et al., 2017), but is equipped with additional attention mechanisms to the factual memory. Consider the $i$th decoding step. We feed the factual memory to an MLP as $\tilde{\mathbf{M}} = \mathrm{MLP}(\mathbf{M})$. We compute attention $\boldsymbol{a}$ over $\tilde{\mathbf{M}}$ with the embedding of the input $\boldsymbol{y}_{i-1}$, and aggregate factual information $\boldsymbol{c}$ for the rows $\boldsymbol{m}_t$ in $\mathbf{M}$:

$$\boldsymbol{a} = \mathrm{softmax}(\tilde{\mathbf{M}} \, \boldsymbol{y}_{i-1}), \quad \boldsymbol{c} = \sum_{k=1}^K a_k \tilde{\boldsymbol{m}}_t^\top$$

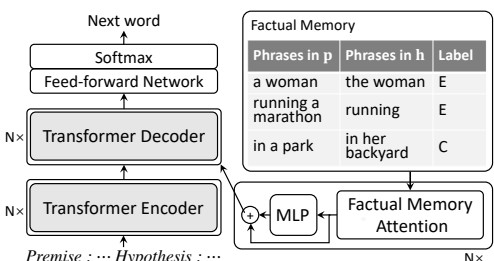

Figure 3: Overview of the model for textual explanation generation.

where $a_k$ is the $k$th element of the vector $\boldsymbol{a}$ and $\hat{\boldsymbol{m}}_t$ is the $k$th row of the matrix $\tilde{\mathbf{M}}$. The factual information $\boldsymbol{c}$ is fed to another layer $\boldsymbol{g}_i = \mathrm{MLP}([\boldsymbol{c}; \boldsymbol{y}_{i-1}]) + \boldsymbol{c}$.

Our Transformer decoder layer starts with self-attention $\tilde{\boldsymbol{q}}_i = \mathrm{SelfAttn}(\boldsymbol{g}_i)$. Then, residual connection and layer normalization are applied as $\boldsymbol{q}_i = \mathrm{LayerNorm}(\tilde{\boldsymbol{q}}_i + \boldsymbol{g}_i)$. A cross-attention mechanism obtains input information by $\boldsymbol{v}_i = \mathrm{CrossAttn}(\boldsymbol{q}_i, \mathbf{H})$, where $\mathbf{H}$ is the representation given by the encoder. $\boldsymbol{v}_i$ is fed to the Transformer's residual connection and layer normalization sub-layer. Multiple Transformer layers as mentioned above are stacked to form a deep architecture. The model is trained by standard cross-entropy loss against the reference explanation as in previous work (Kumar & Talukdar, 2020; Zhao & Vydiswaran, 2021; Narang et al., 2020).

In this way, the model is enhanced with factual information given by our EPR weakly supervised reasoning. Experiments will show that it largely improves the BLEU score by 2 points (§ 4.2), being a new state of the art. This further verifies that our EPR indeed yields meaningful phrasal explanations.

## 4 EXPERIMENTS

### 4.1 DATASETS AND EVALUATION METRICS

The main dataset we used in our experiments is the Stanford Natural Language Inference (SNLI) dataset (Bowman et al., 2015), which consists of 550K training samples, 10K validation samples, and another 10K test samples. Each data sample consists of two sentences (premise and hypothesis) and a sentence-level groundtruth label.[3] For sentence-level NLI prediction, we still use accuracy to evaluate our approach, following previous work (Parikh et al., 2016; Chen et al., 2017; Radford et al., 2018).

To evaluate the phrasal reasoning performance, we need additional human annotation and evaluation metrics, because most previous work only considers sentence-level performance (Feng et al., 2020) and has not performed quantitative phrasal reasoning evaluation. Although Camburu et al. (2018) annotated phrase highlights in their e-SNLI dataset, they are incomplete and do not provide logical relationships. Our annotators selected relevant phrases from two sentences and tagged them with phrasal NLI labels; they also selected and tagged unaligned phrases.

---

[3]A groundtruth label is for a data point, which consists of two sentences. We call it a *sentence-level* label instead of phrasal labels.

Table 2: Main results on the SNLI dataset. [†]Quoted from respective papers. [‡]Obtained from the checkpoint sent by the authors. Other results are obtained by our experiments. GM and AM are the geometric and arithmetic means of the $F$ scores.

| Model | Sent Acc | Reasoning Performance | | | | | | |
|---|---|---|---|---|---|---|---|---|
| | | $F_\text{E}$ | $F_\text{C}$ | $F_\text{N}$ | $F_\text{UP}$ | $F_\text{UH}$ | GM | AM |
| **Human** | – | **84.71** | **71.01** | **55.12** | **82.46** | **61.80** | **70.07** | **71.02** |
| **Non-reasoning** | | | | | | | | |
| Mahabadi et al. (2020)[†] | 85.1 | – | – | – | – | – | – | – |
| LSTM (Wang & Jiang, 2016)[†] | 86.1 | – | – | – | – | – | – | – |
| Transformer (Radford et al., 2018) | 89.9 | – | – | – | – | – | – | – |
| SBERT (Reimers & Gurevych, 2019) | **91.4** | – | – | – | – | – | – | – |
| **Baselines** | | | | | | | | |
| NNL (Feng et al., 2020)[‡] | 79.91 | 62.72 | 17.49 | 1.50 | 66.22 | 0.00 | 0.00 | 29.59 |
| STP | 85.76 | 62.40 | 34.76 | 37.04 | **76.61** | **51.80** | 50.20 | 52.52 |
| GPT-3-Davinci (Brown et al., 2020) | – | 53.75 | **58.00** | 16.12 | 52.24 | 31.08 | 38.23 | 42.24 |
| **Ours** | | | | | | | | |
| EPR (Local, LM unfinetuned) | $76.33_{\pm0.48}$ | $\mathbf{83.11}_{\pm0.29}$ | $38.73_{\pm0.85}$ | $44.63_{\pm0.88}$ | **76.61** | **51.80** | $56.39_{\pm0.43}$ | $58.98_{\pm0.34}$ |
| EPR (Local, LM finetuned) | $79.36_{\pm0.13}$ | $82.44_{\pm0.26}$ | $44.10_{\pm1.32}$ | $\mathbf{44.69}_{\pm3.22}$ | **76.61** | **51.80** | $\mathbf{57.77}_{\pm0.85}$ | $\mathbf{59.93}_{\pm0.67}$ |
| EPR (Concat, LM unfinetuned) | $84.53_{\pm0.19}$ | $73.29_{\pm0.68}$ | $37.95_{\pm1.16}$ | $40.56_{\pm1.10}$ | **76.61** | **51.80** | $53.73_{\pm0.39}$ | $56.04_{\pm0.33}$ |
| EPR (Concat, LM finetuned) | $\mathbf{87.56}_{\pm0.15}$ | $69.91_{\pm1.21}$ | $39.97_{\pm2.12}$ | $43.31_{\pm2.78}$ | **76.61** | **51.80** | $54.46_{\pm1.35}$ | $56.32_{\pm1.13}$ |

We further propose a set of $F$-scores, which are a balanced measure of precision and recall between human annotation and model output for `Entailment`, `Contradiction`, `Neutral`, and `Unaligned` in terms of word indexes. Details of human annotation and evaluation metrics are shown in Appendix B.

The inter-annotator agreement is presented in Table 2 in comparison with model performance (detailed in the next part). Here, we compute the agreement by treating one annotator as the ground truth and another as the system output; the score is averaged among all annotator pairs. As seen, humans generally achieve high agreement with each other, whereas model performance is relatively low. This shows that our task and metrics are well-defined, yet phrasal logical reasoning is a challenging task for machine learning models.

Textual explanation generation was evaluated on the e-SNLI dataset (Camburu et al., 2018), which extends the SNLI dataset with one reference explanation for each training sample, and three reference explanations for each validation or test sample. Each reference explanation comes with highlighted rationales, a set of annotated words in the premise or hypothesis considered as the reason for the explanation annotation. We do not use these highlighted rationales, but enhance the neural model with EPR output for textual explanation generation. We follow previous work (Camburu et al., 2018; Narang et al., 2020), adopting BLEU (Papineni et al., 2002) and SacreBLEU (Post, 2018) scores as the evaluation metrics; they mainly differ in the tokenizer. Camburu et al. (2018) also report low consistency of the third annotated reference, and thus use only two references for evaluation. In our study, we consider both two-reference and three-reference BLEU/SacreBLEU. Appendix A.2 provides additional implementation details of textual explanation generation.

## 4.2 RESULTS

**Phrasal Reasoning Performance.** To the best of our knowledge, phrasal reasoning for NLI was not explicitly evaluated in previous literature. Therefore, we propose plausible extensions to previous studies as our baselines. We consider the study of Neural Natural Logic (NNL, Feng et al., 2020) as the first baseline. It applies an attention mechanism (Parikh et al., 2016), so that each word in the hypothesis is softly aligned with the words in the premise. Then, each word in the hypothesis is predicted with one of the seven natural logic relations proposed by MacCartney & Manning (2009). We consider the maximum attention score as the alignment, and map their seven natural logic relations to our three-category NLI labels: `Equivalence`, `ForwardEntailment` $\mapsto$ `Entailment`; `Negation`, `Alternation` $\mapsto$ `Contradiction`; and `ReverseEntailment`, `Cover`, `Independence` $\mapsto$ `Neutral`.

Table 2 shows that the word-level NNL approach cannot perform meaningful phrasal reasoning, although our metrics have already excluded explicit evaluation of phrases. The low performance is because their soft attention leads to many misalignments, whereas their seven-category logical relations are too fine-grained and cause complications in weakly supervised reasoning. In addition, NNL does not allow unaligned words in the hypothesis, showing that such a model is inadequate for NLI reasoning. By contrast, our EPR model extracts phrases of meaningful semantic units, being an appropriate granularity of logical reasoning. Moreover, we work with three-category NLI labels following the sentence-level NLI task formulation. This actually restricts the model's capacity, forcing the model to perform serious phrasal reasoning.

Table 3: Results of ablation studies on SNLI.

| Model | Features | Sent Acc | Reasoning Performance | | | | | | |
|---|---|---|---|---|---|---|---|---|---|
| | | | $F_E$ | $F_C$ | $F_N$ | $F_{UP}$ | $F_{UH}$ | GM | AM |
| Full model | Local | $76.33_{\pm0.48}$ | $\mathbf{83.11}_{\pm0.29}$ | $\mathbf{38.73}_{\pm0.85}$ | $\mathbf{44.63}_{\pm0.88}$ | **76.61** | **51.80** | $\mathbf{56.39}_{\pm0.43}$ | $\mathbf{58.98}_{\pm0.34}$ |
| | Global | $84.03_{\pm0.12}$ | $70.84_{\pm0.60}$ | $35.12_{\pm0.90}$ | $36.37_{\pm1.52}$ | **76.61** | **51.80** | $51.41_{\pm0.62}$ | $54.15_{\pm0.41}$ |
| | Concat | $84.53_{\pm0.19}$ | $73.29_{\pm0.68}$ | $37.95_{\pm1.16}$ | $40.56_{\pm1.10}$ | **76.61** | **51.80** | $53.73_{\pm0.39}$ | $56.04_{\pm0.33}$ |
| Random chunker | Local | 72.44 | 63.21 | 22.65 | 32.04 | 65.94 | 36.13 | 40.53 | 43.99 |
| | Global | 82.81 | 58.09 | 30.64 | 27.49 | 65.94 | 36.13 | 41.05 | 43.66 |
| | Concat | 83.09 | 58.75 | 32.41 | 31.14 | 65.94 | 36.13 | 42.66 | 44.87 |
| Semantic role labeling | Local | 71.10 | 73.79 | 29.39 | 28.99 | 70.19 | 43.11 | 45.27 | 49.09 |
| | Global | 82.81 | 60.14 | 32.07 | 30.48 | 70.19 | 43.11 | 44.67 | 47.20 |
| | Concat | 83.11 | 61.64 | 31.76 | 28.33 | 70.19 | 43.11 | 44.15 | 47.01 |
| Random alignment | Local | 68.52 | 59.32 | 21.79 | 26.20 | 51.43 | 16.50 | 31.02 | 35.05 |
| | Global | 81.99 | 53.85 | 35.10 | 31.39 | 51.43 | 16.50 | 34.71 | 37.66 |
| | Concat | 82.49 | 57.22 | 34.83 | 30.91 | 51.43 | 16.50 | 34.97 | 38.18 |
| Mean induction | Local | 79.61 | 77.38 | 37.14 | 36.13 | **76.61** | **51.80** | 52.84 | 55.81 |
| | Global | 83.82 | 55.08 | 29.92 | 24.70 | **76.61** | **51.80** | 43.82 | 47.62 |
| | Concat | **84.96** | 57.12 | 31.93 | 31.41 | **76.61** | **51.80** | 46.92 | 49.77 |

In addition, we include another intuitive SBERT-based competing model for comparison. We first apply our own heuristics of phrase detection and alignment (thus, the model will have the same $F_{UP}$ and $F_{UH}$ scores); then, we directly train the phrasal NLI predictor by sentence-level labels. We obtain the sentence NLI prediction by taking `argmax` over Eq. (7). We call this STP (Sentence label Training Phrases). As seen, STP provides some meaningful phrasal reasoning results, because the training can smooth out the noise of phrasal labels, which are directly set as the sentence-level labels. But still, its performance is significantly lower than our EPR model.

We experimented with a baseline of few-shot prompting with GPT-3 (Brown et al., 2020), and the implementation detail is shown in Appendix A.2. We see that GPT-3 is able to provide more or less meaningful reasoning, and surprisingly the contradiction $F$-score is higher than all competing methods. However, the overall mean $F$ scores are much lower. The results show that phrasal reasoning is challenging for pretrained language models, highlighting the importance of our task formulation and the proposed EPR approach even in the prompting era.

Among our EPR variants, we see that EPR with local phrase embeddings achieves the highest reasoning performance, and that EPR with concatenated features achieves a good balance between sentence-level accuracy and reasoning. Our EPR variants were run 5 times with different initialization, and standard deviations are also reported in Table 3. As seen, our improvement compared with the best baseline is around 9.1–10.7 times the standard deviation in mean $F$ scores, which is a large margin. Suppose the $F$ scores are Gaussian distributed,[4] the improvement is also statistically significant ($p$-value $<4.5$e-20 comparing our worse variant with the best competing model by one-sided test).

We further compare our EPR with non-reasoning models (Wang & Jiang, 2016; Radford et al., 2018), which are unable to provide phrasal explanations but may or may not achieve high sentence accuracy. The results show that our phrasal EPR model hurts the sentence-level accuracy by 2–4 points, when the model architecture is controlled. This resonates with traditional symbolic AI approaches (MacCartney & Manning, 2008), where interpretable models may not outperform black-box neural networks. Nevertheless, our sentence-level accuracy is still decent, outperforming a few classic neural models, including fuzzy logic applied to sentence embeddings (Mahabadi et al., 2020).

**Analysis.** We consider several ablated models to verify the effect of every component in our EPR model. (1) Random chunker, which splits the sentence randomly based on the number of chunks detected by our system. (2) Random aligner, which randomly aligns phrases but keeps the number of aligned phrases unchanged. (3) Semantic role labeling, which uses the semantic roles, detected by AllenNLP (Gardner et al., 2018), as the reasoning unit. (4) Mean induction, which induces the sentence NLI label by the geometric mean of phrasal NLI prediction. In addition, we consider local phrase embedding features, global features, and their concatenation for the above model variants. Due to a large number of settings, each variant was run only once; we do not view this as a concern because Table 2 shows a low variance of our approach. Also, the underlying language model is un-finetuned in our ablation study, as it yields slightly lower performance but is much more efficient.

As seen in Table 3, the random chunker and aligner yield poor phrasal reasoning performance, showing that working with meaningful semantic units and their alignments is important to logical reasoning. This also verifies that our word index-based metrics are able to evaluate phrase detection

---

[4]When the score has a low standard deviation, a Gaussian distribution is a reasonable assumption because the probability of exceeding the range of $F$ scores is extremely low.

and alignment in an implicit manner. We further applied semantic role labeling as our reasoning unit. We find its performance is higher than the random chunker but lower than our method. This is because semantic role labeling is verb-centric, and the extracted spans may be incomplete.

Interestingly, local features yield higher reasoning performance, but global and concatenated features yield higher sentence accuracy. This is because global features provide aggregated information of the entire sentence and allow the model to bypass meaningful reasoning. In the variant of the mean induction, for example, the phrasal predictor can simply learn to predict the sentence-level label with global sentence information; then, the mean induction is an ensemble of multiple predictors. In this way, it achieves the highest sentence accuracy (0.43 points higher than our full model with concatenated features), but is 6 points lower in reasoning performance.

This reminds us of the debate between old schools of AI (Chandrasekaran et al., 1988; Boucher & Dienes, 2003; Goel, 2022). Recent deep learning models take the connectionists' view, and generally outperform symbolists' approaches in terms of the ultimate prediction, but they lack expressible explanations. Combining neural and symbolic methods becomes a hot direction in recent AI research (Liang et al., 2017; Dong et al., 2018; Yi et al., 2018). In general, our EPR model with global features achieves high performance in both reasoning and ultimate prediction for the NLI task.

**Results of Textual Explanation Generation.** In this part, we apply EPR's predicted output—phrasal logical relationships—as factual knowledge to textual explanation generation. Most previous studies use the groundtruth sentence-level NLI label and/or highlighted rationales. This requires human annotations, which are resource-consuming to obtain. By contrast, we require no extra human-annotated resources; our factual knowledge is based on our weakly supervised reasoning approach.

Table 4: Textual explanation results on e-SNLI. Previous work uses auxiliary information (L: the groundtruth NLI label; H: human-annotated highlights), but we use neither. [†]Quoted from respective papers. [‡]Evaluated by checkpoints. [∥]Our replication with provided code.

| Model | Info | | BLEU | | SacreBLEU | |
|---|---|---|---|---|---|---|
| | L | H | 2 refs | 3 refs | 2 refs | 3 refs |
| Camburu et al. (2018)[†] | – | – | 27.58 | – | – | – |
| NILE (Kumar & Talukdar, 2020)[∥] | ✓ | – | 28.57 | 37.73 | 32.51 | 41.78 |
| NILE (Kumar & Talukdar, 2020)[‡] | ✓ | – | 28.67 | 37.84 | 32.74 | 42.06 |
| FinetunedWT5$_{220M}$ (Narang et al., 2020)[†] | ✓ | – | – | – | 32.40 | – |
| FinetunedWT5$_{11B}$ (Narang et al., 2020)[†] | ✓ | – | – | – | 33.70 | – |
| LIREx (Zhao & Vydiswaran, 2021)[∥] | ✓ | ✓ | 17.22 | 22.40 | 21.24 | 26.68 |
| Finetune T5$_{60M}$ | – | – | 27.75 | 36.78 | 31.74 | 40.89 |
| + Annotated Highlights$_{64M}$ | ✓ | ✓ | 27.91 | 36.90 | 32.20 | 41.21 |
| + EPR Outputs$_{64M}$ (ours) | – | – | **29.91** | **38.30** | **33.96** | **42.63** |

Table 4 shows our explanation generation performance on e-SNLI. Since evaluation metrics are not consistently used for explanation generation in previous studies, we replicate the approaches when the code or checkpoint is available. For large pretrained models, we quote results from the previous paper (Narang et al., 2020). Their model is called WT5, having 220M or 11B parameters depending on the underlying T5 model. Profoundly, we achieve higher performance with 60M-parameter T5-small, which is 3.3x and 170x smaller in model size than the two WT5 variants.

In addition, we conducted a controlled experiment using the rationale highlights annotated by Camburu et al. (2018) for e-SNLI. It achieves a relatively small increase of 0.2–0.5 BLEU points, whereas our EPR's outputs yield a 2-point improvement. The difference in the performance gains shows that our EPR's phrasal logical relationships provide more valuable information than human-annotated highlights. In general, we achieve a new state of the art on e-SNLI with a small language model, demonstrating the importance of phrasal reasoning in textual explanations.

**Additional Results.** We show additional results as appendices. § C.1: Reasoning performance on the MNLI dataset; § C.2: Error analysis; § C.3: Case studies of our EPR model; and § C.4: Case studies of textual explanation generation.

**Conclusion.** The paper proposes an explainable phrasal reasoning (EPR) model for NLI with neural fuzzy logic, trained in a weakly supervised manner. We further propose an experimental design, including data annotation, evaluation metrics, and plausible baselines. Results show that phrasal reasoning for NLI is a meaningfully defined task, as humans can achieve high agreement. Our EPR achieves decent sentence-level accuracy, but much higher reasoning performance than all competing models. We also achieve a new state-of-the-art performance on e-SNLI textual explanation generation by applying EPR's phrasal logical relationships.

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

## A  IMPLEMENTATION DETAILS

### A.1  PHRASE DETECTION

We present more details about our phrase detection. We use SpaCy[5] to obtain the part-of-speech (POS) tag[6] of every word. SpaCy also tags noun phrases. However, if a noun phrase follows a preposition (with a fine-grained POS tag being IN), we remove it from noun phrases but tag it as a prepositional phrase.

In addition, we extract verbs by the POS tag VERB. A verb may be followed by a particle with the fine-grained POS tag being RP (e.g., *show off*). It is treated as a verb phrase. In order to handle negation, we allow optional AUX NOT before a verb, (e.g., *could not help*). This, however, only counts less than $1\%$ in the dataset, and does not affect our model much.

To capture other potential semantic units, we treat remaining open class words[7] as individual phrases. Finally, the remaining non-content words (in the categories of closed words and others) are discarded (e.g., "there is"). This is appropriate, because they do not represent meaningful semantics or play a

---

[5] `https://spacy.io`

[6] See definitions in `https://spacy.io/usage/linguistic-features`

[7] `https://universaldependencies.org/u/pos/`

Table 5: Our rules for phrase detection. "[]" means the item is optional.

| **Example:** The woman is showing off her blue dog at the playground. | | | |
|---|---|---|---|
| Number | Phrase type | Rule | Extracted phrase(s) |
| 1 | Prepositional phrase | IN + NP | *at the playground* |
| 2 | Noun phrase | NP | *The woman| her blue dog* |
| 3 | Verb phrase | [AUX] + [NOT] + VERB + [RP] | *is showing off* |
| 4 | Others | Other open class words | - |

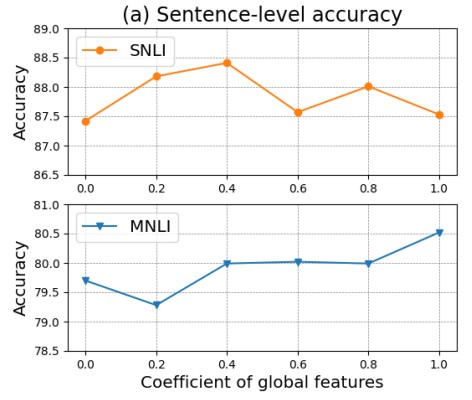
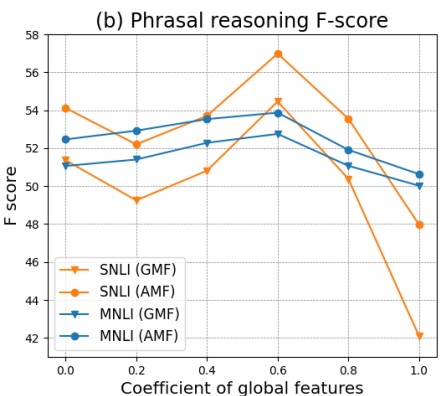

Figure 4: Results of tuning the coefficient of global features.

role in reasoning. Table 5 summarizes all the rules used in our approach. They are executed in order and extracted phrases are exclusive. For example, *the playground* in the phrase *at the playground* will not be treated as a standalone noun phrase, as it is already part of a prepositional phrase.

Empirically, our rule-based approach works well for the NLI dataset, and our logical reasoning is at the granularity of the extracted phrases.

## A.2 SETTINGS

**Details of the EPR Model.** We chose the pretrained model `all-mpnet-base-v2`[8] from the Sentence-BERT study (Reimers & Gurevych, 2019) and obtained 768-dimensional local and global phrase embeddings. Our MLP had the same dimension as the embeddings, i.e., 768D for the local and global variants, or 1536D for the concatenation variant. We chose the coefficient for the global feature in Eq. (1) from a candidate set of $\{0.0, 0.2, 0.4, 0.6, 0.8, 1.0\}$. Figure 4 shows the hyperparameter tuning results on SNLI (mentioned in § 4.2) and MNLI (to be discussed in § C.1). We find that $0.4$ yields the best sentence accuracy in SNLI, and that $1.0$ is the best for MNLI. As our focus is on reasoning, we set the coefficient to be $0.6$, because it yields the highest phrasal reasoning performance and decent sentence-level performance for both experiments and in terms of both geometric mean and arithmetic mean of $F$ scores. The pretrained language model (LM) was either finetuned or un-finetuned during training. Finetuning yields higher performance (Table 2), whereas un-finetuned LM is more efficient for in-depth analyses (Table 3). We trained the model with a batch size of 256. We used Adam (Kingma & Ba, 2015) with a learning rate of 5e-5, $\beta_1$=0.9, $\beta_2$=0.999, learning rate warm up over the first 10 percent of the total steps, and linear decay of the learning rate. The model was trained up to 3 epochs, following the common practice (Dodge et al., 2020). Our main model variants were trained 5 times with different parameter initializations, and we report the mean and standard deviation.

**Details of Textual Explanation Generation.** We used the pretrained T5-small model for finetuning with a batch size of 32. The optimizer was Adam with an initial learning rate of 3e-4, $\beta_1$=0.9, $\beta_2$=0.999, learning rate warm-up for the first 2 epochs, and linear decay of the learning rate up to 10

---

[8]`https://www.sbert.net/docs/pretrained_models.html`

---

The phrasal logical reasoning between two sentences: "Several men helping each other pull in a fishing net." and "There is one man holding the net." is:
1. Entailment: "**pull in a fishing net**" vs. "**holding the net**".
2. Contradiction: "**Several men**" vs. "**one man**".
3. Neutral: **None**.
4. Unaligned premise: "**helping each other**".
5. Unaligned hypothesis: **None**.

The phrasal logical reasoning between two sentences: "An elderly couple are looking at black and white photos displayed on a wall." and "Octogenarians in heavy coats admiring the old photographs that decorated the wall." is:
1. Entailment: "**An elderly couple**" vs. "**Octogenarians**"; "**displayed on a wall**" vs. "**decorated the wall**".
2. Contradiction: **None**.
3. Neutral: "**looking at black and white photos**" vs. "**admiring the old photographs**".
4. Unaligned premise: **None**.
5. Unaligned hypothesis: "**in heavy coats**".

The phrasal logical reasoning between two sentences: "[PREMISE]" and "[HYPOTHESIS]" is:

---

Figure 5: The prompt for phrasal reasoning.

epochs; then we decreased the learning rate to 3e-6 and trained the model until the validation BLEU score did not increase for 2 epochs.

**Details of the Prompting Baseline.** We adopted the GPT-3 (the `text-davinci-003` version with 175B parameters) (Brown et al., 2020) as a prompting baseline to demonstrate large language models (LLMs)' phrasal reasoning ability.

We consider exemplar-based prompting, because it is unlikely for an LLM to output structured reasoning results in a zero-shot manner. Moreover, our examples are chosen to cover all reasoning cases. We also set the temperature of decoding to 0 to obtain deterministic reasoning, following CoT prompting (Wei et al., 2022). Rule-based post-processing was applied to extract slot values. Figure 5 presents the prompt used for phrasal reasoning.

## B    DATA ANNOTATION AND REASONING EVALUATION METRICS

Previous studies have not explicitly evaluated reasoning performance. Typically, they resort to sentence-level classification accuracy (Wang & Jiang, 2016; Mahabadi et al., 2020) or case studies (Parikh et al., 2016; Feng et al., 2020) to demonstrate the effectiveness of their alleged interpretable models, which we believe is inadequate.

Therefore, we annotated a model-agnostic corpus about phrasal logical relationships and developed a set of metrics to evaluate the phrasal reasoning performance quantitatively. The resources are released on our website (Footnote 1) to facilitate future research.

### B.1    DATA ANNOTATION

We annotated the phrases and their logical relationships in a data sample. The annotators were asked to select corresponding phrases from both premise and hypothesis, and label them as either `Entailment`, `Contradiction`, or `Neutral`, with the sentence-level NLI label being given. Annotators could also select a phrase from either a premise or a hypothesis and label it as `Unaligned`. The process can be repeated until all phrases are labeled for a data sample. Figure 6 shows a screenshot of our annotation page. In the left panel, the annotator could select phrases in the two sentences and mark them with NLI labels. The annotator can view a sample's annotated phrases in the right panel and navigate through different samples.

The annotation was performed by three in-lab researchers who are familiar with the NLI task. Our preliminary study shows low agreement when the annotators are unfamiliar with the task; thus it is inappropriate to recruit Mechanical Turks for annotation. We randomly selected 100 samples for annotation, following previous work on the textual explanation for SNLI (Camburu et al., 2018),

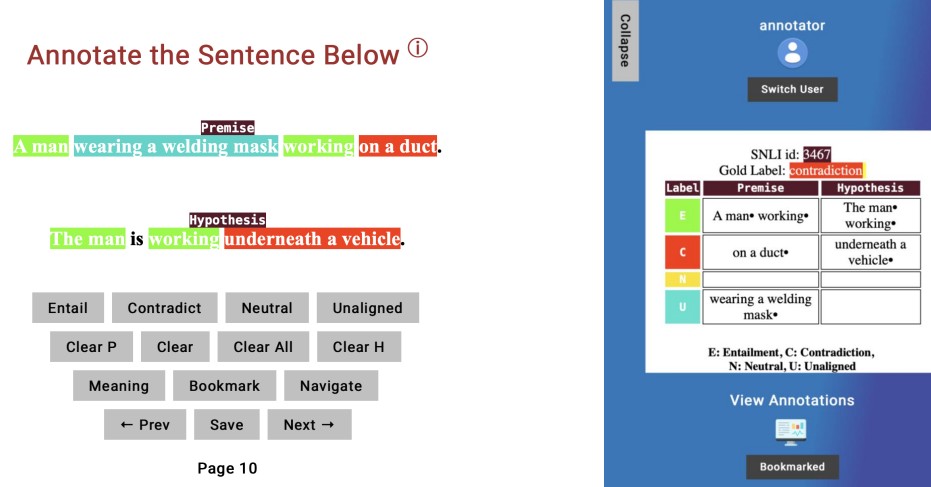

Figure 6: A screenshot of the annotation page.

Table 6: Examples illustrating the proposed metrics, where we consider the `Entailment` category. "|" refers to a phrase segmentation.

| **Example annotation of entailment (in highlight):** | | | | | | | | |
|---|---|---|---|---|---|---|---|---|
| Premise: A kid in red is playing in a garden. | | | | | | | | |
| Hypothesis: A child in red is watching TV in the bedroom. | | | | | | | | |
| # | Example Output | $P_E^{(P)}$ | $P_E^{(H)}$ | $P_E$ | $R_E^{(P)}$ | $R_E^{(H)}$ | $R_E$ | $F_E$ | Explanation |
| 1 | P   in a garden 
 H   in the bedroom | 0 | 0 | 0 | 0 | 0 | 0 | 0 | Although *in* occurs in the annotation, the word indexes are different. The reasoning is wrong. |
| 2 | P   a kid in red 
 H   watching TV | 1 | 0 | 0 | 1 | 0 | 0 | 0 | Mis-matched phrases in hypothesis. The reasoning is wrong. |
| 3 | P   a kid \| in red 
 H   a child \| in red | 1 | 1 | 1 | 1 | 1 | 1 | 1 | All word indexes match the annotation. The reasoning is correct. |

which is adequate to show statistical significance. Since our annotation only concerns data samples, it is agnostic to any machine learning model.

## B.2 EVALUATION METRICS FOR PHRASAL REASONING

We propose a set of $F$-scores in `Entailment`, `Contradiction`, `Neutral`, and `Unaligned` to quantitatively evaluate the phrasal reasoning performance. We first introduce our metric for one data sample and then explain the extension to a corpus.

Consider the `Entailment` category as an example. We first count the number of "hits" (true positives) between the word indexes of model output and annotation. Using word indexes (instead of words) rules out hitting the words in misaligned phrases (Example 1, Table 6). Then, we calculate precision scores for the premise and hypothesis, denoted by $P_E^{(P)}$ and $P_E^{(H)}$, respectively. Their geometric mean $P_E = (P_E^{(P)} P_E^{(H)})^{1/2}$ is considered as the precision for `Entailment`. Here, the geometric mean rules out incorrect reasoning that hits either the premise or hypothesis, but not both (Example 2, Table 6). Further, we compute the recall score $R_E$ in a similar way, and finally obtain the $F$-score by $F_E = \frac{2P_E R_E}{P_E + R_E}$. Likewise, $F_C$ and $F_N$ are calculated for `Contradiction` and `Neutral`. In addition, we compute the $F$-score for unaligned phrases in premise and hypothesis, denoted by $F_{UP}$ and $F_{UH}$, respectively.

When calculating our $F$-scores for a corpus, we use micro-average, i.e., the precision and recall ratios are calculated in the corpus level. This is more stable, especially considering the varying lengths of sentences. Moreover, we compare model output against three annotators and perform an arithmetic average, further reducing the variance caused by ambiguity.

Table 7: Results on MNLI. [†]Quoted from respective papers. [‡]Our replication.

| Model | Sent Acc | Reasoning Performance | | | | | |
|---|---|---|---|---|---|---|---|
| | | $F_E$ | $F_C$ | $F_{UP}$ | $F_{UH}$ | GM | AM |
| **Human** | – | **85.15** | **73.44** | **73.18** | **46.31** | **67.85** | **69.52** |
| **Non-reasoning methods** | | | | | | | |
| Mahabadi et al. (2020)[†] | 73.8 | – | – | – | – | – | – |
| LSTM (Wang et al., 2019)[†] | 72.2 | – | – | – | – | – | – |
| Transformer (Radford et al., 2018) | **82.1** | – | – | – | – | – | – |
| **Reasoning methods** | | | | | | | |
| NNL (Feng et al., 2020)[‡] | 61.28 | 50.33 | 32.00 | 49.78 | 0.00 | 0.00 | 33.03 |
| STP | 75.15 | 55.47 | 51.72 | **64.32** | **37.57** | 51.31 | 52.27 |
| EPR (Concat, LM finetuned) | **79.65**$_{\pm 0.19}$ | **61.76**$_{\pm 0.32}$ | **52.09**$_{\pm 0.41}$ | **64.32** | **37.57** | **52.80**$_{\pm 0.07}$ | **53.93**$_{\pm 0.07}$ |

It should be emphasized that our metrics evaluate phrase detection and alignment in an implicit manner. A poor phrase detector and aligner will result in a low reasoning score (shown in our ablation study), but we do not explicitly calculate phrase detection and alignment accuracy. This helps us cope with the ambiguity of the phrase granularity (Example 3, Table 6).

To summarize, we propose an evaluation framework including data annotation (§ B.1) and evaluation metrics (§ B.2). These are our contributions in formulating the phrasal reasoning task for NLI.

# C  ADDITIONAL RESULTS

## C.1  RESULTS ON MNLI

In this appendix, we provide additional results on the matched section of the MNLI dataset (Williams et al., 2018), which consists of 393K training samples, 10K validation samples, and another 10K test samples. It has the same format as the SNLI dataset, but samples come from multiple domains and are more diverse. We follow § 4.1 and use the same protocol to create the phrasal reasoning annotation for the MNLI dataset based on 100 randomly selected samples. However, we found that MNLI is much noisier than SNLI; particularly, the sentences labeled as Neutral in MNLI share few related phrases. For example, the two sentences do not have much in common in the sample "*Premise: If you still want to join, it might be worked.*" and "*Hypothesis: Your membership is the only way that this could work*". Moreover, the inter-human agreement is low in the Neutral category. Therefore, we believe the corpus quality is less satisfactory for Neutral. To ensure meaningful evaluation, we ignored the evaluation of Neutral in this experiment, although our reasoning approach is not changed. The remaining 60 samples containing Entailment and Contradiction serve as the MNLI phrasal reasoning corpus.

We consider the EPR variant with concatenated local and global features, since the SNLI experiment shows it achieves a good balance between sentence-level accuracy and reasoning. Our models were run 5 times with different initializations.

As seen in Table 7, our EPR approach is again worse than humans, but largely improves the reasoning performance compared with NNL and STP baselines. Its sentence-level prediction is comparable to (although slightly lower than) finetuning Transformers. The results are highly consistent with SNLI experiments, showing the robustness of our approach.

It is important to notice that the EPR model here is trained on MNLI sentence labels, and is not transferred from the SNLI dataset. In our preliminary experiments, we tried transfer learning from SNLI to MNLI and failed to obtain satisfactory performance. We found that our EPR is more prone to the out-of-vocabulary issue (i.e., it does not predict well for the phrases in the new domain), whereas a black-box neural network may learn biased sentence patterns and achieve higher performance in transfer learning.

## C.2  ERROR ANALYSIS

To show how phrasal reasoning affects sentence-level prediction, we perform an error analysis in Table 8. Specifically, we examine the reasoning performance (arithmetic mean of $F$-scores) when the sentence label is correctly and incorrectly predicted on the SNLI dataset. As shown, EPR models

Table 8: Sentence-level prediction count and arithmetic average reasoning performance ($F$-score) when the sentence label is correctly and incorrectly predicted on the SNLI dataset.

| Sentence-level prediction | Count (in percentage) | | Reasoning performance (AMF) | |
|---|---|---|---|---|
| | Local finetuned | Concat finetuned | Local finetuned | Concat finetuned |
| Correct | $75.4_{\pm 1.36}$ | $87.8_{\pm 0.75}$ | $65.71_{\pm 0.83}$ | $58.68_{\pm 0.67}$ |
| Wrong | $24.6_{\pm 1.36}$ | $12.2_{\pm 0.75}$ | $40.74_{\pm 2.01}$ | $37.58_{\pm 3.28}$ |
| Overall | $100.0_{\pm 0.00}$ | $100.0_{\pm 0.00}$ | $59.93_{\pm 0.67}$ | $56.32_{\pm 1.13}$ |

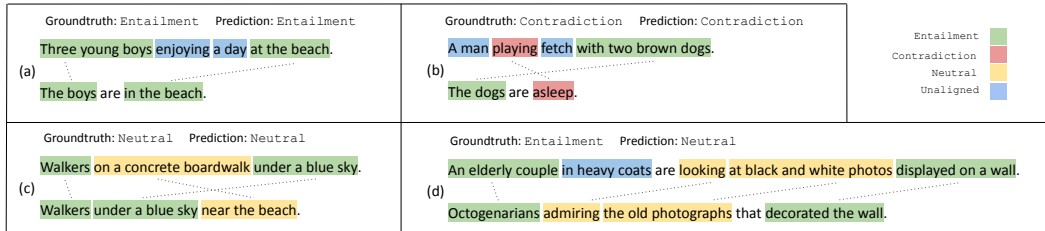

Figure 7: Examples of explainable phrasal reasoning predicted by our EPR model. Words in one color block are detected phrases, a dotted line shows the alignment of two phrases, and the color represents the predicted phrasal NLI label. In Example (d), EPR's prediction suggests the provided label in SNLI is incorrect.

with both local and concatenated features have much higher reasoning performance when sentence labels are correctly predicted than incorrectly predicted. The positive correlation between phrasal reasoning performance and sentence-level accuracy shows our fuzzy logic induction rules indeed make sense.

We also find that the model with local features has a higher reasoning performance than with concatenated features, even when the sentence-level prediction is wrong. This is because the local model is unaware of the context of the sentences. Thus, it must perform strict phrasal reasoning based on the induction rules, even if in this case the reasoning process is imperfect and leads to sentence-level errors.

### C.3 CASE STUDY OF EPR

We present case studies of EPR in Figure 7. Our EPR performs impressive reasoning for the NLI task, which is learned in a weakly supervised manner with only sentence-level labels.

In Example (a), the two sentences are predicted Entailment because *three young boys* entails *the boys* and *at the beach* entails *in the beach*, whereas unaligned phrases *enjoying* and *a day* are allowed in the premise for Entailment. In Example (b), *playing* contradicts *asleep*, and the two sentences are also predicted Contradiction. Likewise, Example (c) is predicted Neutral because the aligned phrases *on a concrete boardwalk* and *near the beach* are neutral.

In our study, we also find several interesting examples where EPR's reasoning provides clues suggesting that the target labels may be incorrect in the SNLI dataset. In Example (d), our model predicts Neutral for *looking* and *admiring*, as well as for *at black and white photos* and *the old photographs*. Thus, the two sentences are predicted Neutral instead of the provided label Entailment. We believe our model's reasoning and prediction are correct, because people looking at something may or may not admire it; a black-and-white photo may or may not be an old photo (as it could be a black-and-white artistic photo).

### C.4 CASE STUDY OF THE TEXTUAL EXPLANATION GENERATION

We conduct another case study to show how EPR's reasoning is used in the textual explanation generation task. As seen in Figure 8, our EPR reasoning yields structured factual tuples: *on a deserted beach* entailing *at the beach*, *Some dogs* contradicting *only one dog*, and *running* unaligned (matched with a special token [EMPTY]). Our explanation generation model attends to these factual tuples, and the heat map shows that our model gives the most attention weights (with an average of

| Input | *Premise : Some dogs are running on a deserted beach.* | | |
| :--- | :--- | :--- | :--- |
| | *Hypothesis : There is only one dog at the beach.* | | |
| **Label** `Contradiction` (not used during our explanation generation) | | | |
| **EPR's Reasoning Output** | | | |
| Premise phrase | Hypothesis phrase | EPR label | Attention score |
| on a deserted beach | at the beach | E | 23.16 |
| Some dogs | only one dog | C | 61.22 |
| running | `[EMPTY]` | E | 15.62 |
| **Output explanation** Some dogs is more than one dog. | | | |
| **Reference explanations:** | | | |
| (1) Some is more than one, therefore there can't be only one dog. | | | |
| (2) Some indicates more than one dog. One dog is not some dogs. | | | |
| (3) Some dogs are not one dog. | | | |

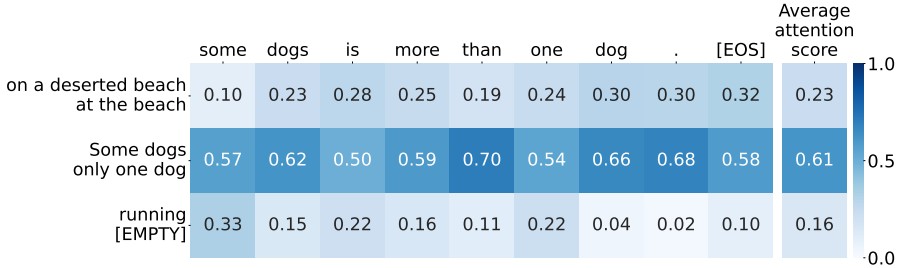

Figure 8: Case study of the textual explanation generation. The heat map shows the step-by-step and average attention weights to the factual tuples (vertical axis).

0.61) to the tuple, *Some dogs* contradicting *only one dog*, to generate the explanation "Some dogs is more than one dog." This example illustrates that the factual tuples given by our EPR model provide meaningful information and can improve textual explanation generation.

## D   LIMITATION AND FUTURE WORK

This paper performs phrase detection and alignment by heuristics. They work well empirically in our experiments, although further improvement is possible (for example, by considering syntactic structures). However, our main focus is neural fuzzy logic for weakly supervised reasoning. This largely differs from previous work based on manually designed lexicons and rules (Hu et al., 2020; Chen et al., 2021).

Our long-term goal is to develop a weakly supervised, end-to-end trained neuro-symbolic system that can extract semantic units and perform reasoning for a given downstream NLP task. This paper is an important milestone toward the long-term goal.

## E   ETHICAL STATEMENTS

Our work involves human annotation of the phrasal logical relationships. Since the research subject here is logic (rather than humans), there are minimal ethical concerns. We nevertheless followed a standard protocol of human evaluation (involving identity protection, and proper compensation), approved by our institional ethics board.

## ACKNOWLEDGMENTS

We thank all reviewers and chairs for their valuable comments. The research is supported in part by the Natural Sciences and Engineering Research Council of Canada (NSERC) under Grant No. RGPIN2020-04465, the Amii Fellow Program, the Canada CIFAR AI Chair Program, a UAHJIC project, a donation from DeepMind, and the Digital Research Alliance of Canada (alliancecan.ca). Atharva Naik contributed to the research as an intern at the University of Alberta through the Mitacs Globalink program.

