# OpenReview forum: "Weakly Supervised Explainable Phrasal Reasoning with Neural Fuzzy Logic"
_ICLR.cc/2023/Conference — ICLR 2023 poster_

### Official Review · Reviewer_vyjc · 2022-10-21

**Confidence:** 3
**Correctness:** 3
**Technical Novelty And Significance:** 2
**Empirical Novelty And Significance:** 3
**Recommendation:** 6

**Clarity, Quality, Novelty And Reproducibility:**

The paper is well written with clear explanations and helpful illustrations. Quality is good and there is some novelty. Codes are available but I didn't check if it is runnable.

**Strength And Weaknesses:**

Strength:
1. The proposed model achieves significant improvement in sentence level accuracy when compared to the previous reasoning model baselines, namely NNL and STP, on the SNLI dataset.
2. Good amount of ablation studies were performed to understand the contributions of each component (e.g. phrase chunking, phrase alignment, fine-tuning etc) .
3. The model is interpretable to some degree in that it can utilize the phrase-level similarity matrix to generate phrase pairs that contribute to the inference results.

Weakness:
1. Even with the sophisticated design, the model performance still lacks behind a non ad-hoc transformer model (Transformer model in Table 2). While the added inductive-bias does add more interpretability, this seems to suggest that training universal large LM is still the best performant choice, as demonstrated by various big LM papers.
2. The proposed phrasal alignment only works when the dataset are well curated and don't use synonyms, I wonder how much it will be applicable in the wild. Specifically, 1) How well does sentence BERT produced embeddings deal with synonyms and different phrasing of the same content? Is there any analysis on this. 2) Real world NL texts are not as concise as in SNLI, there can be added redudant adjectives, disfluencies,  reversed sentence structure. I wonder if alignment will fail for some of these cases.

**Summary Of The Paper:**

The paper proposes a phrase-level NLI model that outperforms previous baselines for NLI on SNLI dataset. The proposed model also has improved interpretability thanks to the phrase-level similarity matrix.

**Summary Of The Review:**

The paper proposes a NLI model that has improved explainability. Evaluation is through, even though results are still not on-par with general purpose transformer models. The proposed method also may have trouble for uncurated data.

---

> ### Author Response · Authors · 2022-11-09
> **Response to Reviewer vyjc**
>
> We thank the reviewer for recognizing our contributions and the significant improvement we achieved compared with other reasoning baseline models.
>
> > Weaknesses 1: “Even with the sophisticated design, the model performance still lacks behind non ad-hoc transformer models.”
>
> We acknowledge that an interpretable model may not always outperform a black-box model. Our work is interested in letting the model provide explicit reasoning results itself. We state in our analysis that there’s a balance between ultimate performance and interpretability within a single model. However, EPR’s phrasal reasoning can improve non ad-hoc transformer models (T5 in our paper) to generate better textual explanations.
>
> As mentioned by other reviewers (VdSJ, 72Kh), slightly lower sentence-level accuracy should not ben considered as a weakness, because our focus is the interpretation. If we want to improve the sentence-level classification performance, we can train a transformer model with a multi-task setting using both sentence-level and phrase-level labels. But multi-task learning is already a well-known technique, so we did not include it in our work.
>
> > Weaknesses 2: “The proposed phrasal alignment only works when the dataset are well curated and don't use synonyms. I wonder if how much it will be applicable in the wild.”
>
> We kindly point out that this is a misunderstanding. We used embedding-based alignment (instead of word match), so it can work with synonyms. In Fig 6d, for example, our method successfully aligned "displayed on a wall" and "decorated the wall". Further, our method can align contradictory phrases (such as playing and asleep), which are not even synonyms.

---

> > ### Author Response · Authors · 2022-11-16
> > **Follow-up message to Reviewer vyjc**
> >
> > Dear reviewer,
> >
> > Thank you very much for your review.. It would be much appreciated if you could check our response, which has clarified both concerns.
> > Should you have any further questions, we’ll be able to provide further clarification in follow-up responses and update our manuscript during the Period-I discussion.
> >
> > Thanks again!
> >
> > -Authors

---

> > > ### Comment · Reviewer_vyjc · 2022-11-27
> > > **Reply to authors's response**
> > >
> > > Thanks for your reply. Regarding point 1, I am OK with the argument that the model trade off some interpretability with performance. Regarding point 2, you did give some examples but do you have statistics of how well it perform? I don't think example is helpful in quantifying how well can the model deal with re-phrasing, sentence re-structuring and etc.

---

> > > > ### Author Response · Authors · 2022-11-28
> > > > **Phrase alignment statistics**
> > > >
> > > > Thanks for suggesting a quantitative analysis of phrase alignment. We report 1) word overlapping ratio in aligned phrases, and 2) the ratio of samples having re-ordered phrases (having a cross/crosses in alignment lines, e.g., Figs 6b and 6c) as follows:
> > > >
> > > > | Category| %word overlap | %re-ordering |
> > > > | --- | --- | --- |
> > > > | Entailment | 55.44 | 16.44 |
> > > > | Contradiction | 18.42 | 14.65 |
> > > > | Neutral | 21.25| 14.98 |
> > > >
> > > >
> > > > We find that the word-overlapping ratio is higher for the Entailment category, whereas the overlapping ratios in the Contraction and Neutral categories are low. This is understandable, because entailed phrases are more likely to use the same words. Nevertheless, the results (especially for N and C) suggest that our approach can perform non-trivial phrase alignment.
> > > >
> > > > For re-ordering, we find that our approach detects around 15% samples with re-ordered alignment. Notice that this is subject to the property of the dataset, and we notice the re-ordering ratio of Entailment is higher than the other categories, which also makes sense because some entailed sentences are expressed by varying the syntactic structure. The results show that our approach works with re-ordering, and indeed, our phrase alignment does not utilize order information, so it should work.
> > > >
> > > > We thank the reviewer again for the additional suggestions. We hope we have alleviated the reviewer’s concern with the above quantitative analyses. We’ll clarify them in the camera-ready version, should the paper be accepted. (Currently, we’re unable to update our manuscript. Thanks for understanding.)

---

### Official Review · Reviewer_72Kh · 2022-10-25

**Confidence:** 4
**Correctness:** 4
**Technical Novelty And Significance:** 3
**Empirical Novelty And Significance:** 4
**Recommendation:** 8

**Clarity, Quality, Novelty And Reproducibility:**

The paper proposes a new task, including an annotated dataset, and a novel, reasonable method for it. The originality is therefore high. Most claims are well supported by the evidence. The provided code and description of experimental setup should lead to good reproducibility. The paper is generally well written, the proposed method is clear. Some minor points:
* captions go above tables according to ICLR guidelines
* non-reasoning baselines are not described, their inclusion is not motivated. Differences in experimental results are not discussed. If you don't discuss something, why include it in the first place?
* while the model by Mahabadi et al. (2020) is substantially different from the proposed model, it also uses fuzzy logic and similar sentence-level classification ideas. The differences should probably be discussed in the related work section, rather than introducing the reference in the middle of the experimental section.
* you are missing a report of whether your data annotation process was approved by an ethical review board

**Strength And Weaknesses:**

Strengths:
* many contributions: a model capable of phrase-level entailment classification, a dataset for evaluation, ablations, multiple use cases
* the paper is well written
* significant interest to the NLP community

Weaknesses:
**Update:** The following weaknesses have been resolved during rebuttal.
* some ablations are missing or too unclear to fully support all claims:
    * You claim that EPR achieves a good balance between sentence-level and phrase-level performance. However, it is not clear in how far the provided non-reasoning baselines are comparable, as they are trained with different underlying pretrained models. If I understand correctly, you are missing a controlled experiment where you use the same underlying LM as your model but finetune it in the "conventional" way, i.e., by plugging a heuristic matching feature layer on top of a siamese encoder, or by training it BERT-style.
    * What is the purpose of your baseline called "Sentence label Training Phrases"? In my understanding it is proposed to show that directly using the sentence-level labels for phrase-level predictions is not suitable. But it doesn't appear to be a controlled experiment, because it uses a different underlying language model? Moreover, how do you obtain sentence-level classification for this model if it is only trained on phrase-level classification?

**Summary Of The Paper:**

The paper proposes a new method for sentence-level natural language inference, which uses intermediate phrase-level entailment annotations to reach to a sentence-level conclusion. First, content phrases are extracted based on a set of heuristics. Secondly, an alignment is found between phrases from the premise and phrases from the hypothesis according to their similarity given by a pretrained text-embedding model. Third, a classification layer predicts class probabilities for each of the alignment pairs. This classification layer is not trained directly from alignment-pair labels, but instead, the predicted probabilities are combined into sentence-level entailment class probabilities, for which labels are available. Hence, phrase-level classification is trained implicitly in a weakly supervised way through sentence-level training.
The model is evaluated on three main tasks: Sentence-level entailment, phrase-level entailment (for which a dataset is created), and explanation generation.
While the proposed model achieves competitive results at the sentence-level, it is the only model capable of phrase-level entailment. For explanation generation, it is demonstrated that inclusion of phrase-level annotations obtained from the model helps to generate better explanations.


**Summary Of The Review:**

The paper makes a number of valuable contributions and has no major weaknesses. I recommend acceptance.

---

> ### Author Response · Authors · 2022-11-09
> **Response to Reviewer 72Kh**
>
> We thank the reviewer for recognizing our contributions and finding our paper interesting.
>
> > Weaknesses 1: “missing a controlled experiment where you use the same underlying LM as your model but finetune it in the "conventional" way”
>
> Thanks for the suggestion. We tried Siamese-structured SBERT and obtained a sentence accuracy of 91.41, higher than fine-tuning the Transformer. We have included the results in Table 3. This is understandable as SBERT is fine-tuned on multiple semantic-related datasets.
>
> However, it should be pointed out that this does not affect the validity of our work, because our main goal is to provide phrase-level interpretation for the NLI task, which cannot be accomplished by SBERT and other previous work. If we do care more about sentence-level accuracy, we can nevertheless apply multi-task learning of SBERT and our EPR model, and use SBERT’s result for sentence prediction.
>
>
> > Weaknesses 2: “What is the purpose of your baseline called "Sentence label Training Phrases"? In my understanding it is proposed to show that directly using the sentence-level labels for phrase-level predictions is not suitable. But it doesn't appear to be a controlled experiment, because it uses a different underlying language model? Moreover, how do you obtain sentence-level classification for this model if it is only trained on phrase-level classification?”
>
> Thanks for asking. The purpose for including this STP baseline is to show that the task is challenging and cannot be adequately solved by easy heuristics. In this baseline, the sentence-level prediction is achieved by taking argmax over Eqn (8). We clarified this in the revision of section 4.2, paragraph 4.
>
> We appreciate that the reviewer emphasizes much on controlled experiments, and so do we. Our controlled experiments are conducted in an extremely rigorous fashion and presented in Table 3.
>
> ### Clarity, Quality, Novelty And Reproducibility:
>
> Thanks again for highlighting our originality and experimental design.
>
> > “captions go above tables according to ICLR guidelines”
>
> Thanks for the suggestion. We changed the location of captions.
>
> > “while the model by Mahabadi et al. (2020) is substantially different from the proposed model, it also uses fuzzy logic and similar sentence-level classification ideas. The differences should probably be discussed in the related work section, rather than introducing the reference in the middle of the experimental section.”
>
> Thanks! In the revision, we discussed in the Related Work section
>
> > “you are missing a report of whether your data annotation process was approved by an ethical review board”
>
> According to our institutional review board, researchers studying logic does not require ethical approval (which is similar to graduate students studying math problems). Here, the subject of the study is logic (not humans). Nevertheless, we do have an umbrella ethics approval for human evaluation, involving identity protection, data storage, and proper compensation. We followed the same protocol here.

---

> > ### Comment · Reviewer_72Kh · 2022-11-09
> > **Thanks for the revision**
> >
> > > "We tried Siamese-structured SBERT and obtained a sentence accuracy of 91.41, higher than fine-tuning the Transformer. We have included the results in Table 3. This is understandable **as SBERT is fine-tuned on multiple semantic-related datasets.**"
> >
> > The goal of my suggestion was to get an honest answer to how much your model loses in terms of sentence-level accuracy on SNLI.
> > If you're saying that SBERT is fine-tuned on multiple semantic-related datasets, do you mean that your accuracy of 91.41 is also obtained from training on SNLI + MNLI (like in the SBERT paper?). If so, why not train it on SNLI only like all your other models? Using SNLI + MNLI for training SBERT makes your model look even worse, don't you agree?
> >
> > Moreover, if you're going to include these numbers in the Table, please contextualize them in the text. Currently, among the non-reasoning baselines, you only explain why Mahabadi et al. makes sense to compare to. What's the purpose of including Wang and Jiang, and Radford et al.? These neither use fuzzy logic nor is the base-LM comparable to yours (i.e., not BERT). If there is no specific reason for including them, just remove them. Use the extra space instead to honestly say in the text that your model loses some sentence-level accuracy compared to the comparable SBERT baseline (which uses heuristic matching features rather than phrase-level reasoning). And that is absolutely fine! It's unreasonable to expect a model that is interpretable at the phrase-level to still retain the same performance at the sentence-level.
> >
> > So in summary, I suggest:
> > * train SBERT on SNLI only
> > * remove Wang and Jiang and Radford et al. from the results table
> > * contextualize your sentence-level results by honestly elaborating on the SBERT results in the paper
> >
> > Of course, if you think my suggestions don't make sense, don't hesitate to push back.
> >
> > > "Thanks for asking. The purpose for including this STP baseline is to show that the task is challenging and cannot be adequately solved by easy heuristics. In this baseline, **the sentence-level prediction is achieved by taking argmax over Eqn (8)**. We clarified this in the revision of section 4.2, paragraph 4."
> >
> > Thank you for clarifying this.
> > You didn't respond to my other question "But it doesn't appear to be a controlled experiment, because it uses a different underlying language model?", but maybe I got confused here myself. Can you confirm that both STP and EPR use the same underlying BERT model?
> >
> > Thank you also for the other modifications. The ICLR guideline asks to include information about any experiments involving humans in the paper, so please add your clarification regarding ethical approval to the appendix.

---

> > > ### Author Response · Authors · 2022-11-10
> > > **Further response and update**
> > >
> > > Hi reviewer,
> > >
> > > Thanks for your prompt reply and your valuable comments! And sorry for the delay because we were running new experiments.
> > >
> > > > Regarding Point 1 of the additional comments:
> > >
> > > Thanks! Our new result, non-reasoning SBERT in Table 2, is indeed fine-tuned on SNLI only. When we said “SBERT is fine-tuned on multiple semantic-related datasets”, we meant that SBERT was fine-tuned (second-stage pre-trained) on multiple semantic-related datasets in the original paper. So this is a fair comparison with our EPR model.
> > >
> > > > Regarding Point 2 of the additional comments:
> > >
> > > We included non-reasoning models (Wang & Jiang, 2016; Radford et al., 2018) because
> > >
> > > 1. They are classic and widely used approaches to the NLI task.
> > > 2. The inclusion of multiple non-reasoning models gives a sense on how different models perform on the dataset. The results show that, although our model is worse in sentence-level accuracy when the model architecture is controlled, its performance is still decent in comparison with other classic models.
> > > 3. Certain readers put more emphasis on established, published baseline models than controlled experiments. However, this is not possible for phrasal reasoning (as it's a new task), so we included additional non-reasoning models to satisfy the curiosity of those readers.
> > >
> > > Since we managed to remain within the page limit, we did not delete those lines, but we cited relevant papers in the text and clearly and honestly explained that our sentence-level accuracy degrades by 2-4 points (compared with Transformer and SBERT).
> > >
> > > > Regarding Point 3 of the additional comments:
> > >
> > > Thanks for catching this! We double-checked our code. We found that our previous STP used BERT as the underlying model, because BERT was our first choice in the development process. We later found SBERT is more effective in phrase detection and alignment, but forgot to update the STP baseline. Now, we have the results in our revised Table 2. Still, STP does not perform well for phrasal reasoning.
> > >
> > > We also included an ethical statement in the appendix. We’re grateful to the reviewer for the constructive suggestions, and we’ve revised our paper accordingly. We’re looking forward to your stronger support!

---

> > > > ### Comment · Reviewer_72Kh · 2022-11-11
> > > > **Major weaknesses have been resolved**
> > > >
> > > > Thanks for additional updates. I don't see any remaining major weaknesses. I think this is a decent paper.

---

> > > > > ### Author Response · Authors · 2022-11-16
> > > > > **Follow-up message to Reviewer 72Kh**
> > > > >
> > > > > We’re truly grateful to the reviewer for the valuable suggestions. They have really improved our paper a lot!

---

### Official Review · Reviewer_K5wq · 2022-11-03

**Confidence:** 4
**Correctness:** 3
**Technical Novelty And Significance:** 3
**Empirical Novelty And Significance:** 3
**Recommendation:** 6

**Clarity, Quality, Novelty And Reproducibility:**

Clarity: the clarity is good in general but some details in the experiment could be more clear.

Quality: This work is of good quality.

Novelty: The idea is novel in the scope of NLI.

Reproducibility: The method is clearly described and the authors claim that the source codes will be open.

**Strength And Weaknesses:**

Strength:

1. The setting of phrase-level NLI is interesting, and is intuitively beneficial for improving the interpretability of neural models.

2. The technical design is sound, and the effectiveness is verified on several datasets.

3. The annotated phrasal logical labels and designed metrics would be useful resources to the community.

Weakness:

1. The presentation of the experimental part is not quite smooth. Both SNLI and e-SNLI datasets are used in the experiments, but it is not clear which datasets the results in Tables 2-3 are based on. A clear indication in the table captions would be better. Besides, as SNLI is mentioned as the main dataset used in the experiments (Section 4.1), my curiosity is raised about how the model will perform in the standard setting of SNLI (i.e., only considering the sentence-level accuracy). I went through the experiments but did find such results. In my understanding, the technique is supposed to be not only useful for the explanation but also generally effective in the standard settings of NLI.

2. Definition of "semantic units" is controversial. Extracting the phrases is the foundation part of this work. According to the details in Appendix A.1, the phrases are extracted based on POS tags. It would be controversial to call those POS-based phrases as "semantic units". In the research line of NLI, there are studies that use semantic roles, which are more likely to be "semantic units". Clarification about this definition would be beneficial.

In addition, as implied in Section 4.2, using different "semantic units" influences the results dramatically. It would also be helpful to see an ablation study by comparing different kinds of language units (e.g., semantic role labels, fact units, etc.), as those units have been more commonly used as fine-grained units in existing studies.

**Summary Of The Paper:**

This paper studies the problem of the explainability of NLI tasks. To address the problem, this paper presents a phrase-based reasoning approach by first detecting the phrases in the given two sentences and aligning those phrases between the sentences. Then, the model is trained to predict the NLI label for the aligned phrases and induce the sentence label by fuzzy logic formulas. With such settings, the model is able to interpret the relationships between the phrases, thus contributing to the explainability of the NLI model.

**Summary Of The Review:**

Overall, this work is interesting in the scope of the NLI field. But the technique is heavily customized to the NLI task, it might have less impact to general audiences. The presentation could also be improved.

---

> ### Author Response · Authors · 2022-11-08
> **Response to Reviewer K5wq**
>
> We thank the reviewer for being interested in our work and supporting our resources for the community.
>
> > Weaknesses 1: “The presentation of the experimental part is not quite smooth. Both SNLI and e-SNLI datasets are used in the experiments, but it is not clear which datasets the results in Tables 2-3 are based on. A clear indication in the table captions would be better.”
>
> Thanks for the suggestion. This is a minor revision, and we have explained the dataset in the captions in our revision.
>
> > Weaknesses 2: “Definition of "semantic units" is controversial. Extracting the phrases is the foundation part of this work. According to the details in Appendix A.1, the phrases are extracted based on POS tags. It would be controversial to call those POS-based phrases as "semantic units". In the research line of NLI, there are studies that use semantic roles, which are more likely to be "semantic units". Clarification about this definition would be beneficial.”
>
> We use the term “semantic units'' to refer to our reasoning ground. We agree that semantic role labeling may be an alternative, but in practice, we find that SRL is too linguistically restricted. Our extraction of semantic units is given by POS-based heuristics. It is easy-to-implement and also generalizable to different tasks (used by a third-party NAACL’22 paper).
>
> > Weaknesses 3: In addition, as implied in Section 4.2, using different "semantic units" influences the results dramatically. It would also be helpful to see an ablation study by comparing different kinds of language units (e.g., semantic role labels, fact units, etc.), as those units have been more commonly used as fine-grained units in existing studies.”
>
> In ablation studies (Table 3), we provided the results for random language units named as “random chunker”. Compared with our well-designed “semantic units”, its reasoning performance decreased by 10-20 F-scores. Further, the NNL approach [Feng et al. 2020] works at the word level and does not achieve a good reasoning performance either (Table 2). These confirm that using meaningful semantic units influences the results dramatically.
>
> We’re grateful for the reviewer's suggestion. We’re currently implementing phrasal reasoning based on semantic role labels. If we obtain results before the author response deadline, we’ll update in the response and the paper. Thanks!

---

> > ### Author Response · Authors · 2022-11-12
> > **Additional response to Reviewer K5wq**
> >
> > Hi reviewer,
> >
> > Thanks again for recommending an alternative phrase detection method using SRL. We have experimented with EPR with semantic role labels. We added this ablated model in the analysis of Section 4.2, and its result is added to Table 3. We adopt the widely used verb-centric (Propbank-style) semantic role labeling; it makes much more sense than random chunker as the performance is much higher. But it shows a performance degradation compared with our full model. We find this because the verb-centric SRL cannot provide a complete span extraction compared with our POS-based extraction.

---

> > > ### Author Response · Authors · 2022-11-16
> > > **Summary of response to Reviewer K5wq**
> > >
> > > Dear reviewer,
> > >
> > > Thanks for recognizing the ”novelty” and “good quality” of this paper.
> > >
> > > The reviewer mainly raised two concerns:
> > > 1. The table captions are not clear enough.
> > > 2. Lack of comparison with different semantic units.
> > >
> > > For concern 1, we revised our paper by explicitly mentioning which dataset is used in each table caption. For concern 2, we conducted an additional experiment, using semantic role labeling as  the semantic units (shown in Table 3). Results show that it is more meaningful than a random chunker, but is much worse than our model.
> > >
> > > As the deadline of period-1 discussion is approaching, we sincerely hope the reviewer could check our revision (especially the new results). We’re happy to make any other changes needed.
> > >
> > > Thank you very much,
> > >
> > > -Authors

---

> > > > ### Comment · Reviewer_K5wq · 2022-11-24
> > > > **Thank you for the response**
> > > >
> > > > Thanks for the detailed response and paper revision. The extra results are helpful.

---

> > > > > ### Author Response · Authors · 2022-11-24
> > > > > **Thanks for reading our response and revision**
> > > > >
> > > > > We greatly thank the reviewer for reading our response and revision, and saying that "The extra results are helpful."
> > > > >
> > > > > Should the reviewer have any further questions, we would still be able to address them in the period-2 discussion. Although we could not update our manuscript at this moment, we will make every effort to clarify in the discussion thread.
> > > > >
> > > > > Thanks!

---

### Official Review · Reviewer_VdSJ · 2022-11-03

**Confidence:** 3
**Correctness:** 3
**Technical Novelty And Significance:** 3
**Empirical Novelty And Significance:** 2
**Recommendation:** 6

**Clarity, Quality, Novelty And Reproducibility:**

The statement of “addressing logical reasoning for the NLI task” is not justified as logical reasoning for the NLI task is not properly defined. It is not clear what the goal of this work is, what the problem is.

It is not clear what the term “phrasal reasoning” refers to. It seems like it is simply the task of NLI on sub-sequences. It should be further motivated why this is considered reasoning.

Question: where in Figure 2.c is the Fuzzy Logic step? Is the “normalize” box doing the fuzzy logic? I would make the paper more clear to include that step in the Figure 2.c.


**Strength And Weaknesses:**

**Strengths**:

This work proposes a novel technique to do NLI at a sub-sentence level. It is interesting to analyze this behavior in order to better understand how the model behaves.

In addition, this work proposes a heuristic approach relying on SpaCy tagger to create the sub-sentence chunks.

Eventually, the set of fuzzy logical rules is also a contribution of this work.

**question** :  can the authors report how the quality of the phrase definition influences the performance of the sentence-BERT alignment which in turn influences the overall NLI accuracy?

**Weaknesses**:

From table 2 it looks like the proposed method is not better than the Transformer baseline on the overall sentence accuracy. It should be mentioned that this work proposes an interpretable way of performing the task of NLI by essentially doing NLI on sub-phrases but this is at the cost of a weaker overall performance.

It is not clear if this work proposes a new task (sub-phrases NLI) or a new method for doing NLI. In both cases it is not clearly shown that predicting “phrases” NLI labels is beneficial for the NLI task. Maybe the set of fuzzy logical rules are not optimally designed for the model to take advantage of the “phrases” labels? It is very possible that large models rely on other statistical features not captured by our human judgment of what may or may not help performing NLI.


**Summary Of The Paper:**

This paper proposes an interpretable method for Natural Language Inference: first, the premise and hypothesis sentences are splitted into “phrases” (or sub-sentences) with a custom heuristic method, then a pre-trained sentence-BERT is used with cosine-similarity to align phrases from the premise to phrases from the hypothesis, then a Transformer network predicts a NLI label (Entailment / Contradiction / Neutral) for each pair of aligned phrases. Finally, a set of fuzzy logical rules combine those predictions to infer the overall sentence pair NLI label.

Experiments show that the phrase alignments and their NLI labels agree with human judgment. However it seems like the overall NLI prediction is not beating a simpler Transformer baseline.


**Summary Of The Review:**

This paper represents a lot of work but its motivation is not clearly presented. It is interesting to analyze sub-phrases NLI alignment but results do not show a strong improvement over a transformer baseline on the NLI task. The motivation of this work should be clearly mentioned at the beginning of the paper.

---

> ### Author Response · Authors · 2022-11-08
> **Response to Reviewer VdSJ (1/2)**
>
> We thank the reviewer for saying that “This work proposes a novel technique to do NLI at a sub-sentence level“ and recognizing our contributions to defining fuzzy logical rules for the task.
>
> > Question: "Can the authors report how the quality of the phrase definition influences the performance of the sentence-BERT alignment which in turn influences the overall NLI accuracy?"
>
> The random chunker in our ablation study (second row of Table 3) has reported the performance of EPR when we randomly chunk sentences (the number of chunks is based on our phrase detection module). Despite its simplicity, this rigorous ablation study shows that the quality of the phrase does influence both the phrasal reasoning F-scores and the sentence-level accuracy, and that our approach can effectively detect phrases as the unit of reasoning.
>
> > Weaknesses 1: "From table 2 it looks like the proposed method is not better than the Transformer baseline on the overall sentence accuracy. It should be mentioned that this work proposes an interpretable way of performing the task of NLI by essentially doing NLI on sub-phrases but this is at the cost of weaker overall performance.
>
> We acknowledge that our proposed method does not perform better than the Transformer model on sentence-level accuracy, as our approach focuses on the interpretability of a black-box neural model (the transformer-based model in our paper). However, our performance is only 2% lower than fine-tuning Transformer, and it is even better than some other black-box neural models.
>
> We mentioned in our last paragraph of our analysis where our approach attempts to balance connectionists’ and symbolists’ approaches: a trade-off between performance and interpretability (as traditional symbolic AI approaches are known to achieve lower performance than neural networks).
>
> > Weaknesses 2 "It is not clear if this work proposes a new task (sub-phrases NLI) or a new method for doing NLI. In both cases it is not clearly shown that predicting “phrases” NLI labels is beneficial for the NLI task. "
>
> Thanks for asking. Indeed, our contributions include both a new sub-task of NLI and a new method. Predicting phrasal relationships to NLI is beneficial because
> 1.  It provides interpretability to neural networks, which is a hot topic in the neural network literature (Liu et al., 2018; Amizadeh et al., 2020; Garcez et al., 2020).
> 2. We are able to improve the e-SNLI performance (an established textual explanation generation task) by 2 BLEU points with our EPR reasoning results.
>
> As clearly stated in the last paragraph of the Introduction, our contributions are an entire pipeline of phrasal logical reasoning, including task formulation, a novel approach, data annotation and evaluation, as well as application to a downstream task (namely, textual explanation generation). The completeness and comprehensiveness should be considered as strengths of our work, instead of weaknesses.
>
> > Weaknesses 3:  "Maybe the set of fuzzy logical rules are not optimally designed for the model to take advantage of the “phrases” labels? It is very possible that large models rely on other statistical features not captured by our human judgment of what may or may not help performing NLI."
>
> Thanks for the comments. We agree that the fuzzy logical rules are not 100% accurate (and no machine learning models can achieve 100% accuracy anyway).
>
> In fact, we have modified and extended traditional fuzzy logic, which uses product for conjunction (Footnote 3). In our development, we find this gives a very small number, causing difficulty in learning. Therefore, we designed the Entailment Rule as geometric mean, which can project the scale back.
>
> To the best of our knowledge, there were no existing studies applying fuzzy logic rules to NLI. Nor could any prior work predict phrase-level logic relationships in a weakly supervised manner.
>
> Therefore, our work is an important breakthrough, even if our accuracy is not 100% yet. Should the reviewer provide more concrete suggestions about the design of fuzzy logic rules, we would be very grateful to the reviewer and try these approaches. Thanks!

---

> > ### Author Response · Authors · 2022-11-08
> > **Response to Reviewer VdSJ (2/2)**
> >
> > > Clarity 1: "The statement of “addressing logical reasoning for the NLI task” is not justified as logical reasoning for the NLI task is not properly defined. It is not clear what the goal of this work is, what the problem is."
> >
> > Our goal is to explain NLI predictions with phrasal logical relationships between the premise and hypothesis, as shown in Figure 1. We have clarified this in the introduction section of the revision.
> >
> > > Clarity 2: "It is not clear what the term “phrasal reasoning” refers to. It seems like it is simply the task of NLI on sub-sequences. It should be further motivated why this is considered reasoning."
> >
> > According to Wikipedia, “Reason is the capacity of consciously applying logic by drawing conclusions from new or existing information”. In other words, we can think of reasoning as intermediate thinking steps to draw a conclusion.
> >
> > In our paper, the phrasal logical relationships are the intermediate thinking steps to determine sentence-level relationships. Thus, this should be considered as reasoning.
> >
> > Our phrasal reasoning refers to reasoning about phrasal logical relationships. We’ve clarified this in the revision of Introduction.
> >
> > > Clarity 3: "where in Figure 2.c is the Fuzzy Logic step? Is the “normalize” box doing the fuzzy logic? I would make the paper more clear to include that step in the Figure 2.c. "
> >
> > Thanks for asking. The “normalize” box is to normalize fuzzy logic scores as probabilities, i.e., the Z under Eqn (8). We’ve clarified this by annotating “fuzzy logic” in the figure.

---

> > > ### Comment · Reviewer_VdSJ · 2022-11-09
> > > **response to authors**
> > >
> > > Thanks for your quick clarifications and answers. The revised version of the paper is more clear now. Score updated to 6.
> > >
> > > ---
> > >
> > > > We mentioned in our last paragraph of our analysis where our approach attempts to balance connectionists’ and symbolists’ approaches.
> > >
> > > In the last paragraph of your analysis, you indeed mention this trade-off by saying the following:
> > > > Recent deep learning models take the connectionists’ view, and generally outperform symbolists’ approaches in terms of the ultimate prediction, but they lack expressible explanations. Combining neural and symbolic methods becomes a hot direction in recent AI research (Liang et al., 2017; Dong et al., 2018; Yi et al., 2018). In general, our EPR model with global features achieves high performance in both reasoning and ultimate prediction for the NLI task.
> > >
> > > Which is not the same thing as saying that (as previously suggested), the proposed interpretable NLI method is at the cost of slightly weaker overall NLI performance. Please note that the small performance miss is definitely not a weakness. Interpretability is always important, especially for deep neural nets. The suggestion is to explicitly say in the analysis that the method loses a little performance for the benefit of greater interpretability. Not explicitly saying it is a lack of transparency.
> > >
> > > ---
> > >
> > > > As clearly stated in the last paragraph of the Introduction, our contributions are an entire pipeline of phrasal logical reasoning, including task formulation, a novel approach, data annotation and evaluation, as well as application to a downstream task (namely, textual explanation generation). The completeness and comprehensiveness should be considered as strengths of our work, instead of weaknesses.
> > >
> > > As clearly stated in the list of strengths mentioned for this paper, the completeness and comprehensiveness was never considered a weakness. The concern raised by "weakness #2" was not about the completeness and comprehensiveness of the work. It was about the transferability of phrasal NLI to the original NLI task due to the performance miss-match raised earlier. However, after reading the revised version, the goal of this paper is "to explain NLI predictions" and not necessarily to perform better on the original NLI task, thus the transferability to better NLI performance becomes irrelevant and "weakness #2" can be ignored now that the goal of the work is explicitly stated in the introduction.
> > >
> > > ---
> > >
> > > > our work is an important breakthrough, even if our accuracy is not 100% yet. Should the reviewer provide more concrete suggestions about the design of fuzzy logic rules, we would be very grateful to the reviewer and try these approaches.
> > >
> > > Again, the weakness was not about not achieving 100% accuracy, that would actually be suspicious as well. As clearly mentioned in the list of strength for this work, the list of fuzzy logic rules is a major contribution. The intention of "weakness 3" is not to propose any additional rules.
> > > The concrete suggestion raised with "weakness #3" is to discuss in the analysis section the very interesting results of this work, namely: since adding human interpretability features to a neural model does not clearly outperforms the transformer baseline, it must be that the traditional transformer rely on other statistical biases in the dataset which are very likely non-interpretable. This is an interesting thing to say in this work, and weakness #3 is simply asking to discuss more about that.

---

> > > > ### Author Response · Authors · 2022-11-10
> > > > **Further response and update**
> > > >
> > > > Hi Reviewer,
> > > >
> > > > Thanks for the insightful comments.
> > > >
> > > > > Additional comment 1:
> > > >
> > > > Thank you for the suggestion. We updated our manuscript and clearly mentioned that our EPR model leads to accuracy degradation of a few points (Page 8).
> > > >
> > > > > Additional comment 2:
> > > >
> > > > Thanks for understanding. This weakness is resolved.
> > > >
> > > > > Additional comment 3:
> > > >
> > > > The reviewer raised a very interesting point: why interpretable models lead to performance degradation.
> > > >
> > > > We have two hypotheses:
> > > >
> > > > 1. Our logical induction rules are strict, and it may lead to error accumulation: a wrong phrase prediction will likely cause a wrong sentence-level prediction.
> > > >
> > > > 2. Indeed, a black-box neural network may learn spurious patterns, whereas our EPR model cannot do so easily. During our development, we tried transfer learning from SNLI to MNLI and our EPR does not achieve satisfactory results. Through a case study, we found that our EPR is more prone to the out-of-vocabulary issue (i.e., it does not predict well for the phrases in the new domain), whereas a black-box neural network may learn biased sentence patterns. Due to the low performance, we didn’t pursue this direction, and we did not have quantitative results at this moment. We nevertheless put a discussion in Appendix C.1. We are grateful for the reviewer’s comment as it suggests a new direction from our study: transferability/systematic generalization of neuro-symbolic models.
> > > >
> > > >
> > > > We thank the reviewer again and hope we have addressed all the concerns.

---

### Decision · Program_Chairs · 2023-01-20

**Decision:**

Accept: poster

**Justification For Why Not Higher Score:**

The proposed method is quite complex, includes multiple disconnected steps and hence might be brittle. The interpretability also comes at a slight cost to accuracy.

**Justification For Why Not Lower Score:**

This paper does produce good empirical results and interpretable NLI is of interest to the community.

**Metareview: Summary, Strengths And Weaknesses:**

This paper proposes a method for interpretable natural language inference. The pipeline includes using an automated method for splitting the text into reasoning units (phrases), using pre-trained BERT embeddings to align the extracted phrases from the premise to those within the hypothesis, using a transformer network to predict the NLI label (entailment/contradiction/neutral) for each pair, and finally aggregating these using fuzzy logic into the final NLI label.

This method appears to work well empirically on a number of datasets, and its interpretability is verified by human judgement. Producing interpretable NLI results is also an important problem to address. Saying this, this method does not outperform a simple transformer baseline (albeit the transformer is a black box), and hence interpretability comes at a slight cost.

**Note From Pc:**

if the above contains the word "oral" or "spotlight" please see: "oral" presentation means -> notable-top-5% and "spotlight" means -> notable-top-25%. As stated in our emails, we are disassociating presentation type from AC recommendations